# Recent Advances in Nanomaterial-Based Biosensors for Pesticide Detection in Foods

**DOI:** 10.3390/bios12080572

**Published:** 2022-07-27

**Authors:** Ana Carolina de Morais Mirres, Brenno Enrique Pereira de Matos da Silva, Leticia Tessaro, Diego Galvan, Jelmir Craveiro de Andrade, Adriano Aquino, Nirav Joshi, Carlos Adam Conte-Junior

**Affiliations:** 1Department of Natural and Technological Sciences, University of Grande Rio (UNIGRANRIO), Duque de Caxias, Rio de Janeiro 25071-202, Brazil; anacarolinamm4@gmail.com (A.C.d.M.M.); brennopi@hotmail.com (B.E.P.d.M.d.S.); 2Laboratory of Advanced Analysis in Biochemistry and Molecular Biology (LAABBM), Department of Biochemistry, Federal University of Rio de Janeiro (UFRJ), Cidade Universitária, Rio de Janeiro 21941-909, Brazil; leticiatessaro@pos.iq.ufrj.br (L.T.); diegogalvann@iq.ufrj.br (D.G.); jelmirandrade@pos.iq.ufrj.br (J.C.d.A.); aquinolp@gmail.com (A.A.); nirav.joshi@ifsc.usp.br (N.J.); 3Center for Food Analysis (NAL), Technological Development Support Laboratory (LADETEC), Federal University of Rio de Janeiro (UFRJ), Cidade Universitária, Rio de Janeiro 21941-909, Brazil; 4Nanotechnology Network, Carlos Chagas Filho Research Support Foundation of the State of Rio de Janeiro (FAPERJ), Rio de Janeiro 20020-000, Brazil; 5Analytical and Molecular Laboratorial Center (CLAn), Institute of Chemistry (IQ), Federal University of Rio de Janeiro (UFRJ), Cidade Universitária, Rio de Janeiro 21941-909, Brazil; 6Graduation Program of Chemistry (PGQu), Institute of Chemistry (IQ), Federal University of Rio de Janeiro (UFRJ), Cidade Universitária, Rio de Janeiro 21941-909, Brazil

**Keywords:** nanobiosensor, agrochemical, food contamination, pesticide contamination, pesticide residues, food safety

## Abstract

Biosensors are a simple, low-cost, and reliable way to detect pesticides in food matrices to ensure consumer food safety. This systematic review lists which nanomaterials, biorecognition materials, transduction methods, pesticides, and foods have recently been studied with biosensors associated with analytical performance. A systematic search was performed in the Scopus (*n* = 388), Web of Science (*n* = 790), and Science Direct (*n* = 181) databases over the period 2016–2021. After checking the eligibility criteria, 57 articles were considered in this study. The most common use of nanomaterials (NMs) in these selected studies is noble metals in isolation, such as gold and silver, with 8.47% and 6.68%, respectively, followed by carbon-based NMs, with 20.34%, and nanohybrids, with 47.45%, which combine two or more NMs, uniting unique properties of each material involved, especially the noble metals. Regarding the types of transducers, the most used were electrochemical, fluorescent, and colorimetric, representing 71.18%, 13.55%, and 8.47%, respectively. The sensitivity of the biosensor is directly connected to the choice of NM and transducer. All biosensors developed in the selected investigations had a limit of detection (LODs) lower than the Codex Alimentarius maximum residue limit and were efficient in detecting pesticides in food. The pesticides malathion, chlorpyrifos, and paraoxon have received the greatest attention for their effects on various food matrices, primarily fruits, vegetables, and their derivatives. Finally, we discuss studies that used biosensor detection systems devices and those that could detect multi-residues in the field as a low-cost and rapid technique, particularly in areas with limited resources.

## 1. Introduction

Pesticides control various pests and increase crop productivity and efficiency [1]. However, the overuse of pesticides in agriculture is linked to damage to the environment and consumers’ health because of the high pesticide residues in food [2,3]. For example, in humans, the most concerning are organophosphate and carbamate pesticides, which inhibit the enzyme acetylcholinesterase (AChE), responsible for several biochemical reactions [4]. To detect these pollutants in low concentrations, sensitive analytical procedures are required [5]. Thus, among the most commonly utilized techniques are high-performance liquid chromatography (HPLC) [6], or liquid chromatography (LC-MS), and gas chromatography (GC-MS) coupled with mass spectrometry [5,6]. However, several systems can be used such as gas chromatography (GC) with electron capture detection, flame ionization detection, or nitrogen-phosphorus detection, mass spectrometry and/or liquid chromatography (LC) with ultraviolet, diode array, fluorescence, or electrochemical detection and mass spectrometry. These methods used expensive equipment and extensive pretreatment operations, and also required the use of highly skilled professionals [7]. In this context, biosensors can be used as a viable alternative that supplements traditional analytical approaches by simplifying or removing the sample preparation phase [7].

Briefly, a biosensor is composed of a receptor, a transducer, and a biorecognition substance that detects certain target molecules in the medium. Figure 1 depicts the research that has been published on the detection of pesticides, which are increasing each year (dotted line), by utilizing various approaches. The continuous line depicts the detection application of biosensors, emphasizing the growing use of this approach to monitor these chemicals.

There is still the option of incorporating nanomaterials into the devices, enhancing selectivity and sensitivity, making analysis more efficient, easy, fast, and economical, with improved accuracy, robustness, and field deployment capacity [8,9]. Gold nanoparticles (AuNPs) [10], silver nanoparticles (AgNPs) [11], and carbon nanotubes (CNTs) are among the most promising nanomaterials that can be used in the building of biosensors [12], aside from nanohybrids, which are composed of two or more linked nanomaterials [13].

The detecting process occurs in transducers which generate an analytical signal confirming the detection [14]. Transducers can be classified into numerous types, the most common of which are electrochemical, optical, piezoelectric, and calorimetric [15]. Colorimetric detection has the advantage of requiring no optical tools to visualize/confirm detection. The downside of this transduction is that it is less sensitive than others, such as electrochemical or fluorescence. However, nanomaterials and biofunctionalization elements have circumvented this barrier [16]. Fluorescents also have various advantages in measuring analyte concentration, but the main drawback is the need for optical equipment to measure the fluorescence created during detection [17].

Choosing the type of transducer used in the biosensor is a critical task, especially when nanomaterials are present, because it directly influences the detection sensitivity of the analyte target [18]. Figure 2 depicts a schematic of the stages required to construct a biosensor, including the selection of nanomaterials used to boost sensitivity.

In light of the exposition, we highlight the difference between our study and those already published in the literature [19,20,21]. This work involved a systematic and thorough review of the recent literature from 2016 to 2021 in the main scientific databases. In this literature, biosensors were based on nano-materials because they improve performance [22], mostly in terms of sensitivity, which is important for these analytes that are found at the trace level in food matrices. For this purpose, a survey of (i) nanomaterials, (ii) biorecognition materials, and (iii) transducers most used in this situation was carried out, as well as which showed superior detection performance. Additionally, which (iv) pesticides have these biosensors detected in (v) food samples?

## 2. Systematic Research Methods

Based on pre-established criteria, this systematic review chose publications from three scientific databases, analyzing abstracts, keywords, and titles. Articles that did not include biosensors that detect pesticides in food using nanomaterials were omitted. As a result, the statement flow diagram for Preferred Reporting Items for Systematic Review and Meta-Analyses (PRISMA) was used [23].

### 2.1. Focus Questions

The population, intervention, comparison, and outcome (PICO) strategy is supported by the focus question. The following were the research questions: (1) How do nanomaterial-based biosensors detect pesticides in food? (2) What methodologies and nanomaterials and transducers were used? (3) Which provides the best analytical performance? (4) What types of food and pesticide samples do these biosensors analyze?

### 2.2. Search Strategy and Selection Criteria

This search strategy was carried out on 22 June 2021 through searches in three databases (Web of Science, Scopus, and Science Direct) for articles published between the years 2016 to 2021. The search was restricted to research articles in English, through for the title, abstract, and keywords, and the choice of criteria strings followed were: (i) identification of keywords considering the research question and (ii) use of the Boolean operators “AND”, “OR” and “*”. As shown in Search Components (SCn).
SC1 = biosensor *SC2 = pesticide * OR agrochemical *****

After retrieving the results from the search component, the Boolean operator “AND” was used to combine SC1 and SC2. Meanwhile, the asterisk functioned as the wildcard operator, which looked for words starting with the root/stem of the word preceding the operator.

Table 1 describes the inclusion/exclusion criteria used in this systematic review. The writers began by selecting abstracts, keywords, and article names that had been identified separately. In this initial screening, the article was eliminated if it did not study the relationship between biosensors and nanomaterials for detecting pesticides in food. Furthermore, some articles were omitted because they did not satisfy the study’s objectives.

### 2.3. Data Extraction

Data extraction and quality assessment were performed independently by three reviewers. If there was doubt about the study’s eligibility, the authors did not exclude it and decided only after a full-text reading. The authors have screened the full-text publications and decided whether these met the eligibility criteria. Data were extracted from selected articles, including data on the type of (i) nanomaterial, (ii) transducer, (iii) biorecognition material, (iv) pesticide, (v) analytical figures of merit (AFOM), and (vi) food samples.

## 3. Dataset Visual Approaches

Figure 3 shows the PRISMA flow diagram with the results of the systematic search. Tables were created to demonstrate some similarities and differences among the 57 papers chosen for quantitative synthesis. Because detection methods and nanomaterials differ, we will explore how this affects detection and which varieties show higher sensitivity. In addition, we also evaluate the influence of nanomaterial and transducer types on the limit of detection (LOD) of biosensors in the following sections of this systematic review.

### 3.1. Influence of the Type of Nanomaterial on the Sensitivity of the Biosensor to Pesticides

Nanomaterials are commonly used in biosensors because they significantly improve performance, allowing faster, more efficient, and affordable detection. The performance improvement is due to their unique optical and electrical properties, which generate high contact surface-to-volume ratio, high electrical conductivity, catalytic activity, biocompatibility, and can be easily modified with functional groups [22]; such as gold nanoparticles (AuNPs), silver nanoparticles (AgNPs), silver nanowires (AgNWs), gold nanorods (AuNRs), gold nanostars (AuNSs), carbon nanotubes (CNT), copper nanowires (CuNWs), and multi-wall carbon nanotubes (MWCNTs). Apart from pure materials, some hybrid nanostructures have also been investigated. Table 2 shows the most often used nanomaterials in biosensor applications based on publications selected for inclusion.

While selecting articles to compose the systematic review, different types of nanomaterials associated with varying types of biorecognition materials and other detection systems were observed. Some systems are applied from colorimetric to electrochemical detection, depending on the properties of the nanomaterial. SWCNTs and MWCNTs, for example, exhibit excellent thermal conductivities, whereas Graphene has a higher surface area than CNTs. Graphene oxide (GO), whose electronic conductivity is less, and Carbon-based quantum dots have unique characteristics that make them extraordinary materials for diverse applications, such as photoluminescence properties, biocompatibility, and low toxicity. Metal-based nanomaterials Au or Ag, for example, have a high surface area and have the excellent adsorption ability of small molecules. These properties are associated with the low detection limits obtained with biosensors. Following are listed the main nanomaterials used to manufacture biosensors to detect pesticides [79].

#### 3.1.1. Gold Nanomaterials (AuNMs)

AuNMs are commonly used to fabricate biosensors because of their unique optical and electronic characteristics. An essential feature of this material is the absorption of intense and well-defined surface plasmon resonance (SPR) signals in the visible region. SPR is a phenomenon that results from the excitation of metal electrons through electromagnetic radiation, generating a variation in the dielectric constant (gold is susceptible to this dielectric constant). Consequently, there is a variation in the light reflection from the surface of the metal in liquid [10,80,81,82]. These nanoparticles increase the apparent mass of the immobilized analytes through electronic coupling between the localized surface plasmon of AuNPs, increasing the chance of detection, thus improving the sensor’s sensitivity [10,81,82,83,84,85]. Figure 4 depicts typical TEM images of Au nanoparticles of various shapes and sizes that span a wide size range, from 15 to 190 nm, and whose size distributions are generally narrow and used in a variety of applications such as sensors and biosensors.

AuNM-based biosensors showed low LOD, as demonstrated by Zhao et al. in the detection of methomyl which reported a LOD value of 81 ng L^−1^ for a biosensor with AChE immobilization using the substrate mercaptomethamidophos and AuNMs together with a glassy carbon electrode (GCE). In another study, eleven organophosphate insecticides with LODs ranging from 19 to 77 ng L^−1^ were found. These LOD values are obtained due to the increased surface area induced by the nanoscale effect and the excellent conductivity of AuNMs. The immobilizing mercaptomethamidophos stayed on the electrode surface via Au–S bonds; the strong contact between AChE and mercaptomethamidophos ensures that AChE can be stuck to the electrode surface in high amounts [24].

In another study, Lin et al. developed an electrochemical biosensor that, through Au–S bonds, detected chlorpyrifos in apple and pak choi samples with a LOD of 36 ng L^−1^. This approach was employed to increase the response signal. When attached to an aptamer probe, the diol sulfite group at the end of a ferrocyanide probe is immobilized on the electrode surface of AuNMs and two-dimensional molybdenum compounds through the development of a double strand. The ribbon is melted when the aptamer-containing chlorpyrifos is inserted and the ferrocyanide approaches the electrode, amplifying the signal. Because the electrode has high electrical conductivity and a large surface area, pesticide detection is performed satisfactorily [26]. Hou et al. created a biosensor with a LOD of 70 × 10^−3^ ng L^−1^ that detects chlorpyrifos in Chinese cabbage and lettuce samples using AuNMs and chlorpyrifos antibodies. As an analyte competitor, the biosensor employs modified AuNMs, bovine serum albumin (BSA), and horseradish peroxidase. As the concentration of chlorpyrifos increases, so does the impedimetric signal [27].

An optical biosensor was developed and utilized to detect isocarbophos in cabbage, tea, and peach samples, with a LOD of 2.48 × 10^3^ ng L^−1^. An aptamer-based lateral flow biosensor (LFB) approach was used, in which the aptamers desorb from the surface of AuNMs when interacting with the pesticide and are subsequently bound to cysteamine, which turns red with intensity proportional to the concentration of isocarbophos [28].

For isocarbophos, chlorpyrifos, carbamate, and methomyl, studies using AuNMs showed good specificity and sensitivity ranging with LOD from 70 × 10^−3^ [27] to 2.48 × 10^3^ ng L^−1^ [28]. The high sensitivity of AuNMs for these tests and the ability to visualize the results for colorimetric detectors without analytical tools are considerable advances in the biosensor scenario.

#### 3.1.2. Silver Nanomaterials (AgNMs)

AgNMs are also commonly used in the manufacture of biosensors. The benefits of adopting this nanomaterial are its large surface area, high electron transport efficiency, and commercial availability [33,34]. Figure 5 shows some silver nanomaterials of different shapes and sizes.

Wang and Liu used guanine-rich DNA (G-DNA) and AgNPs doped with Terbium(III) (Tb^3+^) to create a fluorescence biosensor for detecting organophosphorus pesticides in apples. Lanthanides and AgNPs work well together because their ordering with guanine-rich DNA results in the transfer of energy from the DNA to Tb^3+^ when exposed to UV light, improving the fluorescence of DNA-Tb-AgNPs. As a result, a detection limit of 34 ng L^−1^ was achieved [32]. In another study by Bala et al. an optical biosensor with AgNPs was developed to detect malathion in apples with a LOD of 5 × 10^−4^ nM by employing malathion-specific aptamers that react with AgNPs to give a colorimetric response. The AgNPs show yellow in the absence of the pesticide due to particle interaction with the aptamer and peptide link, and orange in the presence of malathion [34].

In order to detect paraoxon residues in chive and Chinese cabbage, Zheng et al. developed an electrochemical biosensor based on AChE immobilization. AgNPs were chosen because of their advantage in detecting thiocholine (TCh) at low voltage without extra modifications. The electrode uses chitosan as a binder to immobilize AChE, and as a result, they obtained a low LOD of 4 × 10^3^ ng L^−1^ [33]. At the same time, Turan et al. employed silver nanoparticles in wire form, known as silver nanowires (AgNWs), in the composition of an amperometric biosensor. AgNWs were used on a polymer-coated surface to increase the charge transfer rate. This butyrylcholinesterase (BChE) immobilization platform also includes a modified graphite electrode that achieved a LOD of 212 nM to detect paraoxon in milk [35].

The use of AgNMs in biosensor construction indicated good sensitivity, with a detection range of 5 × 10^−4^ nM with AgNPs in apple samples for the malathion pesticide [34] and 212 nM with AgNWs in milk samples for the paraoxon pesticide [35]. Besides, AgNMs have higher visibility in colorimetric biosensors and have the advantage of being a less expensive choice.

#### 3.1.3. Carbon Nanotubes (CNTs)

CNTs are the most commonly employed nanomaterials to generate nanohybrids, either alone or in combination with other nanomaterials. CNTs are carbon allotropes that feature a cylindrical-shaped lattice of carbon atoms in one or more layers with open or closed ends. Its usage in the creation of biosensors is widespread because it has features that make it particularly sensitive when exposed to biomolecules [88]. CNTs have a high surface area, strong mechanical strength, outstanding electrical conductivity, electrochemical stability in aqueous and non-aqueous solutions, and high thermal conductivity. Furthermore, they have distinct inherent optical features like near-infrared photoluminescence and significant resonance Raman scattering. Simultaneously, the electrochemical properties have reactive assemblies installed on the outer surface, which can increase electron transfer [88,89]. Figure 6 shows some silver nanomaterials of different carbon nanotubes (CNT) at different magnifications in thin sheets.

It is possible to obtain different carbon structures, such as graphene, nanosheets, and mesoporous spheres, including nanotubes [42]. Due to the advantages of these carbon-based nanomaterials, biosensors have been widely applied to other detections, especially in the biomedical field [91] or for detecting different pesticide classes [12]. Kaur et al. reported an electrochemical biosensor based on AChE immobilization in multi-walled carbon nanotubes (MWCNTs) functionalized and wrapped in poly(3,4-ethylenedioxythiophene) to detect chlorpyrifos-methyl in lettuce with a LOD of 1 ng L^−1^ [51]. At the same time, another biosensor was created by Kaur et al. which used electrochemical detection and AChE enzymes for malathion detection in lettuce with a LOD of 1 × 10^−6^ nM [29]. Different segments were also proposed, such as the use of rabbit antibodies to create specific polyclonal antibodies that undergo an immunoreaction with malathion, distinguishing this study from others that have already been published. Furthermore, polystyrene sulfonate (PEDOT) and multi-walled carbon nanotubes (MWCNTs) combine to generate a nanocomposite that immobilizes antibodies on its surface [51].

Different CNTs strategies were checked in electrochemical biosensors with different food and pesticides. Han et al. created an electrochemical biosensor for paraoxon detection in spinach juice samples, achieving a LOD of 3 × 10^−6^ nmol L^−1^. The biosensor was created by fusing CNT with amino acid ionic liquid, which increased the electrochemical activity of the mineralized cell (M-Cell), which was then fused with organophosphate hydrolase as the biorecognition material [57]. Chen et al. created a chlorpyrifos-detecting electrochemical biosensor. The herbicide and AChE were immobilized on MWCNTs to detect it in lettuce and cabbage samples, with an outstanding LOD sensitivity of 50 ng L^−1^ [50]. Another electrochemical biosensor, AChE-based, built to detect the organophosphorus insecticide malathion and methyl parathion in Chinese cabbage samples, had a LOD of 3.11 × 10^−4^ and 1.88 × 10^−4^ ng L^−1^, respectively. This study shows how single-wall carbon nanotubes (SWCNTs) can be employed with Prussian Blue to operate as a low-potential redox mediator and electron transfer facilitator [58].

In contrast, Lin et al. created a fluorescent biosensor with a “turn-on” mechanism to detect acetamiprid waste in pre-treated cabbage leaf samples with a LOD of 0.7 nM. This biosensor also featured Mn-doped ZnS and Quantum Dots linked to acetamiprid aptamers in addition to MWCNTs. Because of strong stacking interactions with double bond groups, the broad wavelength range of the absorption spectra of carbon nanostructures permits the transfer of energy from fluorescence resonance covering the spectrum of fluorophores. As a result, MWCNTs are used as fluorescence inhibitors, activating only when the analyte is present. This biosensor can potentially be helpful for on-site visual testing [56].

CNTs as biosensor constituents showed excellent sensitivity and detection time for different class pesticides, with limits ranging from 1 × 10^−6^ nM [29] to 50 ng L^−1^ [50]. The great advantage of using CNTs is a large surface-to-volume ratio, good mechanical and chemical stability, and a high electron transfer rate.

#### 3.1.4. Graphene and Graphene Oxide (rGO)

Graphene is a hexagonal network of sp2 hybridized carbon atoms covalently bonded. The treatment of graphite with strong oxidants adds epoxy groups, hydroxyl groups, and carboxyl groups to its structure, thus producing graphene oxide, which is reduced, generating reduced graphene oxide (rGO) [92]. The properties of rGO include better conductivity concerning graphene oxide, better dispersion in solvents due to the presence of functional groups, ease of control of electrical performance and solubility of rGO, ease of manufacture, and relatively low cost. Overall, graphene and reduced graphene oxide have excellent properties such as exceptionally high stiffness and mechanical strength, attractive for the construction of flexible devices, excellent electrical conductivity, high optical transparency, and good biocompatibility; due to this, they have potential application in portable electronic devices [93]. This section discusses some studies included in this SR that use this nanomaterial.

An electrochemical biosensor based on graphene membrane with magnetic nanoparticles (Fe_3_O_4_) with inhibition of AChE was developed by Wang et al. to detect chlorpyrifos. The membrane has properties such as a large specific surface area and high electron transfer, providing a better detection and effective immobilization of AChE. The large number of active sites provided by the nanocomposite favors catalysis reactions and makes the environment suitable for improving the AChE reaction. The biosensor obtained excellent sensitivity with a LOD of 20 ng L^−1^ and has multi-use capacity for this just to perform the immersion of the device in pralidoxime chloride solution, recovering 90% of its original activity [59]. While Dong et al. developed an electrochemical biosensor for detecting malathion and methyl parathion in Chinese cabbage based on the inhibition of AChE, a scheme is presented in Figure 7. A film using three-dimensional (3D) rGO was prepared with nickel foam (NF) combined with AuNPS (AuNPs/rGO/NF). This combination provides a large surface area for AChE adsorption and excellent electron transfer due to the synergistic effect of AuNPs and rGO, besides being environmentally friendly. This study showed a satisfactory sensitivity LOD of 2.17 × 10^−2^ ng L^−1^ [67] when compared to the biosensor developed by Li et al., and a LOD of 3.9 × 10^2^ ng L^−1^ for the malathion [70]. The study by Li et al., used rGO modified with tetraethylenepentamine (TEPA) and copper nanowires (CuNWs) to improve the conductivity and load capacity of the electrode for AChE. The copper metal has catalytic and electrochemical properties supplied, and its union to other nanomaterials, forming the so-called rGO-TEPA/Cu NWs nanocomposites generate a network structure that improves the specific surface of the sensor and increases detection sensitivity [70].

Graphene and rGO nanomaterials proved to be an excellent alternative for pesticide detection in food samples. The LOD of the biosensors varied from 2.17 × 10^−2^ [67] to 3.9 × 10^2^ ng L^−1^ [70], a difference in the order of 10^4^. This discrepancy can be explained by the fact that the more sensitive biosensor also uses AuNPs with unique properties that have proven to be ideal for biosensors.

#### 3.1.5. Quantum Dots (QDs)

Quantum Dots (QDs) are colloidal nanocrystalline semiconductor crystals that exhibit continuous absorption spectra which range in length from the ultraviolet to the visible, depending on the particle size. The optical and spectroscopic features of QDs, such as fluorescence, give these materials advantages over traditional fluorophores in various applications, with popularity in the biomedical field. Furthermore, its unique properties include highly effective catalytic activity, broad electron excitation, and size-adjustable emission wavelength; that is, the loss of light absorption capacity is small, and it has high photochemical stability [94,95,96]. These characteristics make the use of promising QDs in biosensors that mainly use light excitation and can be formed from various materials.

Yang et al. developed a biosensor with Ir nanorods with Cadmium Sulfide Quantum Dots (Ir NRs@CdSQDs) to detect paraoxon in cabbage pakchoi, and lettuce, see Figure 8. The device uses electrochemiluminescence, with Ir NRs as an anodic emitter and CdS QDs as a cathode emitter. In the presence of organophosphate pesticides, there is the immobilization of the biocomposite of AChE and choline oxidase (ChOx), and the reaction between enzymes generates H_2_O_2,_ which increases the cathode signal and decreases the anodic. This biosensor has a high LOD of 1.67 × 10^−3^ nM, RSD <5%, and long-term satisfactory stability [76]. Another device with Cadmium Telluride Quantum Dots (CdTe QDs) was developed by Korram et al., which, due to its properties, allowed the sensitive detection of traces without spectral interference. The fluorescent biosensor is based on the AChE inhibition mechanism for multiple optical detections for organophosphate pesticides such as paraoxon, dichlorvos, malathion, and triazophos in apple and tomato juice. This detection method is simple, consisting of a single step, mixing the sample with AChE and the solution containing AChE, CHOx, and QDs. The sensitivity obtained a LOD of 1.62 × 10^−6^ to 0.23 nM, presenting better sensitivity to paraoxon and less to malathion [65]. A fluorescent probe to detect acetamiprid using ZnS:Mn QDs together with MWCNTs and target-specific acetamiprid aptamer in cabbage samples was developed by Lin and coauthors. The Mn-doped ZnS QDs with acetamiprid aptamer has excellent fluorescent properties, a characteristic responsible for the probe’s sensitivity, and lower toxicity than the CdTe QDs popularly used. The advantage of this biosensor is the detection without complex pre-treatments of the sample, obtaining a LOD of 0.7 nM, and high selectivity generated by the aptamer [56].

The biosensors included in this SR used QDs of different materials: ZnS:Mn QDs, Ir NRs@CdSQDs, and CdTe QDs in their compositions. The main advantage was the high sensitivity in detecting organophosphate pesticides, ranging from 1.62 × 10^−6^ to 0.23 nM. As a highlight, they presented the possibility of multiple detections of pesticides and detection without complex pre-treatments by the study by Korram et al. [65].

#### 3.1.6. Titanium Nanomaterials (TiNMs)

Titanium nanomaterials are frequently used in the form of titanium dioxide (TiO_2_), which feature a variety of nanostructures. TiO_2_ NMs can reach a large surface area and have unique chemical, physical and electronic properties; in addition, they have the advantages of being non-toxic, biocompatible, and photocorrosion-free. These NMs can be prepared on a large scale in mild temperatures and conditions, so they are easy to manufacturing and have low cost. In addition, they can be doped by other elements, which results in increased or decreased conductivity, depending on the coupled material. TiO_2_ NMs have often been proposed as an interface for the enzyme immobilization of biomolecules due to the preservation of biocatalytic activity and being a good electron donor in a reaction between biomolecules and analyte that occurs in biosensors [97,98,99,100]. Titanium dioxide nanoparticles (TiO_2_NP) were used in conjunction with serine, histamine, and glutamic acid amino acids, forming a nanoenzyme capable of hydrolyzing organophosphates. This nanoenzyme was used to construct an electrochemical biosensor to detect methyl paraoxon, methyl parathion, and ethyl paraoxon in lettuce samples. Nanoenzymatic composites have higher hydrolysis activity than TiO_2_ or pure amino acids, so the coexistence of the three amino acids with TiO_2_ gave the highest catalytic activity for the hydrolysis of OPs with a LOD of 220 to 260 nM [61]. On the other hand, the biosensor produced by Hu et al. uses vitreous carbon electrodes with TiO_2_NP and chitosan, forming a TiO_2_-based sensor sol-gel carrier. The mechanism of action was the inhibition of the AChE for detecting dichlorvos in cabbage juice samples. This biosensor obtained a LOD of 0.23 nM [60], approximately 100 times more sensitive than the nanoenzymatic biosensor developed by Qiu and coauthors. In contrast, Li et al. used titanium in the shape of a thin 2 nM film covered with a 20 nM gold layer in a microcantilever device to detect profenofos in Chinese chives. Detection was performed using aptamers as biorecognition materials since aptamers can conjugate with various types of target molecules with high affinity and specificity. This biosensor obtained a LOD of 1.3 × 10^3^ ng L^−1^, surpassing other detection methods based on aptamers [62].

The studies selected SR that used biosensors with TiNMs that were efficient for detecting organophosphate pesticides such as dichlorvos, profenofos, methyl paraoxon, methyl parathion, and ethyl paraoxon in different food samples. The work developed by Qiu et al. was able to perform multiple detections of three pesticides with nanoenzymatic composites in a single detection. Finally, the biosensors developed showed excellent sensitivity, with LODs ranging from 0.23 nM [60] to 260 nM [61] and 1.3 × 10^3^ ng L^−1^ [62], values below that required by current legislation (S1), presenting a great potential for monitoring pesticides in food.

#### 3.1.7. Hybrid Nanostructures

Nanohybrids are nanomaterial compositions that result in unique features that can improve the sensitivity of biosensors. Nanomaterial nanohybrid has the advantage of combining the properties of two metals [13], which enhances its benefits such as amplification of SPR signals, surface area, biocompatibility, electron transfer [36], and the combination with other nanomaterials such as carbon [43], silicon [38], etc.

In this sense, Rahmani et al. created an electrochemical biosensor without enzymes using a bimetallic nanocluster of silver and gold wrapped in the BSA protein. This biosensor detected methyl parathion with a LOD of 8.2 nmol L^−1^ in apple, cabbage, spinach, and lettuce. It has been demonstrated that bimetallic nanoclusters of Ag and Au have increased optical properties due to the transfer of load across metals, as well as improved plasma stability and biocompatibility. Because of their ultra-small size and particular electrical structure, they interact better with biological systems [37]. Xu et al. developed an electrochemical biosensor based on AChE immobilization and used it to detect the pesticides methyl parathion, malathion, and chlorpyrifos in cabbage juice, with LODs of 3.04 × 10^−3^ ng L^−1^, 1.96 × 10^−3^ ng L^−1^, and 2.06 × 10^−3^ ng L^−1^, respectively. The synergy of vertical nitrogen-doped single-walled carbon nanotubes (VNSWCNTs) and self-assembled AuNPs resulted in good electron transport. This technique produced exceptional sensitivity and the ability to identify pesticide residues numerous times [40]. Another biosensor for pesticide detection in cabbage juice was proposed by Cui et al. [38]. The built biosensor with nanohybrids composed of TiO_2_, chitosan, gold nanorods (AuNRs), and mesoporous silicon dioxide had a detection limit of 1.3 nM for fenthion in juice samples (see Figure 9A). This nanomaterial has good properties because it makes good contact with the electrode surface and has a mesoporous nanostructure, which provides mechanical strength, increased AChE charging efficiency, a large specific surface area, low cost, no toxicity, good thermal and chemical stability, and excellent biocompatibility. The chitosan polymer is especially advantageous since it immobilizes the enzyme, resulting in a high surface area and biocompatible nanostructure [38]. Ma et al. reported an electrochemical biosensor of AChE with LODs ranging from 8.6 × 10^−6^ to 7.1 × 10^−5^ nM for detecting malathion, methyl parathion, and chlorpyrifos in potato and corn. The biosensor is built with an N-doped carbon-core shell and a bimetallic core (Pt and Pd), providing significant benefits such as outstanding electrochemical properties. Furthermore, carbon-based porous structures have excellent features such as good electrical conductivity, a large specific surface area, size distribution, and stability [43].

Yang et al. employed an acetamiprid-binding aptamer (ABA) covalently coupled to AuNPs in a duplex configuration. The fluorescent biosensor was equipped with upconversion nanoparticles (UCNPs) and a double-stranded DNA-functionalized AuNP probe for detecting acetamiprid in celery leaves and Chinese green tea samples, with a LOD of 0.36 nM [39]. Another AChE-based fluorescence biosensor found ethyl parathion in apple and orange juice samples. Because of the previously mentioned SPR capabilities of the Ag and Au metals, a bimetallic nanohybrid Au@Ag Nanoclusters (NC) was employed, which allowed detection with the naked eye (LOD of 2.40 × 10^−3^ nM) even at low pesticide concentrations. Furthermore, peptide-based techniques for the NC synthesis of metals avoid using hazardous reductants by functioning as a reducing agent and capping agent [41].

Another article discusses two electrochemical biosensors using enzymes for detecting malathion in pear samples. One biosensor used mesoporous hollow carbon spheres (MHCs), while the other used MHCs core-shell structures with magnetic nanoparticles denoted Fe_3_O_4_@MHCs, as shown in Figure 9B, yielding LOD values of 14.8 ng L^−1^ and 18.2 ng L^−1^, respectively. Carbon nanoparticles exhibit a variety of properties, as previously stated. Mesoporous spheres created from these nanomaterials have properties that influence detection, such as a large surface area, porosity, and electrical conductivity. While Fe_3_O_4_ magnetic nanoparticles have advantages such as large surface area, high chemical stability, and low toxicity, they also function as electron-conducting pathways, facilitating electron transfer between redox systems and mass electrode materials and remaining stable under conditions such as high temperature, high pressure, and large pH variations. Combining these magnetic nanoparticles with the carbon coating inhibits nanoparticle aggregation, provides a broad support area for subsequent changes, and has highly linked porous architectures and well-protected magnetic components [42]. In addition, a surface-enhanced Raman scattering (SERS) effect biosensor using DNA aptamers and Ag@Au bimetallic nanoparticles developed by Lu et al. detected profenofos, acetamiprid, and carbendazim pesticides, which resulted in LODs of 2.1 ng L^−1^, 4.6 ng L^−1^, 6.1 ng L^−1^, respectively, in rice and apple samples. This combination of metallic nanoparticles provides outstanding Raman response activity and dispersibility [36].

The nanohybrids demonstrated excellent sensing capability in terms of sensitivity and selectivity. They have the advantage of combining the properties of two metals, enhancing positive effects such as SPR signal amplification, surface area, biocompatibility, and electron transport. As a result of these discoveries, this method is a viable candidate for developing new biosensors for detecting pesticides in food. Figure 9 depicts a selection of the images discussed in this section.

### 3.2. Effect of the Transducer Type on the Limit of Detection

Transducers have an essential role in the detection of a chosen target substance. The main function is to convert an analytical signal into an informational reading signal. Due to this fact, many fronts of studies have evaluated different approaches; optical (colorimetric and fluorescence), electrochemical, amperometric, and SERS transducers are among the most used. Table 3 summarizes the types of transducers used and their analytical performance in detecting pesticides in food based on selected articles according to the inclusion criteria.

#### 3.2.1. Colorimetric Transducer

Colorimetric transducers have advantages over traditional methods: they are less expensive, lighter, portable, have a smaller sample size, and require less instrumentation. These transducers can be tailored to a specific analyte or a broad range of analytes for qualitative or quantitative research [101]. Devices can be produced using simple inkjet printing and connected with appropriate user-friendly data gathering and analysis apps, which are even accessible for smartphones. Some of the difficulties reported by different writers who used this sort of transducer include printing repeatability, imaging acquisition, trouble distinguishing individual components of a combination, and stability/shelf-life [102].

Some uses of these transducers have been investigated. Bala et al. reported a favorable outcome in their work on the experimental assay for detecting malathion in apple samples utilizing aptamer as a biorecognition material and AgNPs as a nanomaterial. The acquired analytical values were a LOD of 5 × 10^−4^ nM and an RSD of 2.98%. This biosensor’s sensitivity was comparable to that of other transducers that require additional equipment to observe the results, as shown in Figure 10A [34]. Another use of colorimetric transducers was reported by Chen et al. to detect parathion in samples of pear, cabbage, and rice. The researchers employed a bimetallic nanomaterial composed of AuNPs and PtNPs functionalized with complementary DNA molecules. Compared to other colorimetric methods, the test findings demonstrate strong detection sensitivity, with a LOD of 2 ng L^−1^ and RSDs of 5.19%, 9.81%, and 15.75% for pear, cabbage, and rice, respectively [45]. At the same time, Liu et al. used colorimetric transduction to detect the pesticide isocarbophos in cabbage, peach, and tea using AuNPs functionalized with aptamers. The biosensor had a LOD of 2.48 × 10^3^ ng L^−1^ and an RSD of 2.37 to 7.13%, showing outstanding performance in sensitivity and precision during the experiment compared to conventional colorimetric transducer-based techniques [28] (see Figure 10B). Acetochlor and fenpropathrin insecticides were detected in pak choi, cabbage, and lettuce samples utilizing antibodies as biorecognition material. The sensitivity was 6.3 × 10^2^ ng L^−1^ for acetochlor pesticide and 2.4 × 10^2^ ng L^−1^ for fenpropathrin pesticide, with an RSD of 3.30%, validating the biosensor’s high sensitivity and precision. [75].

The colorimetric transducers showed LOD ranging from 5 × 10^−4^ nM [34] to 6.3 × 10^4^ ng L^−1^ [75], employing several detection targets and primarily AuNPs as a nanomaterial. This type of colorimetric transduction was employed in 5 of the 57 experiments. Figure 10 depicts the evolution of some colorimetric biosensors mentioned in this section.

#### 3.2.2. Electrochemical Transducer

Electrochemical sensing systems have several advantages in fabrication and selecting tools or materials for their creation due to the simplest and cheapest techniques. Other advantages include portability and compatibility with various materials and biological components, suitable sensing, and stable results. However, this type of sensor has some drawbacks, such as a lack of detecting sensitivity for many specific materials [31,103].

The electrochemical transducer was the most commonly utilized pesticide detection tool. Many writers combined electrochemical transducers with optical and infrared transducers. The most commonly utilized nanomaterial was metal nanoparticle probes such as gold and iron, which give good stability during detection, while AChE was the most commonly employed target detection for biosensor manufacturing [31,49,103].

Amperometric transducers are another type of electrochemical sensing device. They can be classed as independent sensing system devices due to a number of advantages, including lower costs and a simpler fabrication platform than other electrochemical sensing systems [104]. With a LOD of 212 nM, an amperometric transducer successfully detected parathion insecticides in milk samples. The use of conjugated polymer poly(TTBO) and the nanomaterial AgNWs improved the assay procedure’s sensitivity to enhance sensitivity [35]. Turan et al. claim that the biosensor can identify pesticides in various samples, including environmental, clinical, food quality, and safety control [35].

These studies show that this transducer has excellent sensitivities, ranging from 3 × 10^−6^ nmol L^−1^ [57] and 1.0 × 10^−6^ nM [51]. Furthermore, this type of transducer was the most used in studies in 40/57 selected due to speed, simplicity, and robustness.

#### 3.2.3. Fluorescence Transducer

The fluorescence technique has advantages such as high sensitivity and selectivity, in addition to performing detection quickly compared to methods that use absorbance, depending on the surrounding environment. Compared to absorbance methods, this transducer type has a sensitivity that can reach 100 times higher due to specific interactions between fluorophores and surface plasmons in nanomaterials of metallic structures. Its high selectivity is due to the spectra obtained from the specifically excited compounds [105,106].

This transducer demonstrated outstanding sensitivity in the range of 1.62 × 10^−6^ [65] to 5 × 10^4^ ng L^−1^ [73]. It was present in 8/57 research, being the second most used transducer, with the major constituents functionalized in nanomaterials such as AChE and aptamers. With an emphasis on the work of Korram et al., with the creation of the most sensitive biosensor in this class, quantum dots were utilized to detect paraoxon, dichlorvos, malathion, and triazophos with a LOD of 1.62 × 10^−6^ nM in orange and tomato juice samples [65]. This technique, as previously said, has a high sensitivity; nonetheless, it requires optical instrumentation to visualize detection data.

#### 3.2.4. Microcantilever-Array Sensor

Microcantilever-array sensing technology originated from atomic force microscopy and was developed in the 1990s, providing a new analytical platform with high throughput, selectivity and sensitivity, label-free detection, and low cost. The microcantilever device is based on aptamers with optical detection and is capable of real-time pesticide detection, and can function in various vacuum, air, and liquid conditions. The microfabrication category is mostly employed as force sensors to map the topography of a surface using techniques such as scanning force microscopy (SFM) or atomic force microscopy (AFM). More details are provided by Lang et al. [62].

Thin-film titanium covered with a gold layer as the nanomaterial functionalized with a profenofos-specific aptamer as the detection target was used by Lang et al. to develop a microcantilever-array sensor. This device displayed outstanding sensitivity detection to profenofos in Chinese chives sample with a LOD of 1.3 × 10^3^ ng L^−1^. A unique interaction between profenofos and the aptamer created a substantial deflection on the microcantilever. Compared to other transducers, this transducer was the most accurate [62].

#### 3.2.5. Piezoelectric Transducer

Piezoelectric transducers are often realized with a cantilever sensing system, implying that the platform transmits data on pressure, acceleration, force, and displacement. Some works used that sensor system in conjunction with microelectromechanical system devices to demonstrate microcantilever beam transducing; however, the method becomes more difficult as more connecting devices are added [47]. The piezoresistive cantilever sensors are most commonly used in aqueous media, one of their primary detection advantages. Another advantage of being a small-sized sensor is portability, and they can be quite sensitive provided the right material and application are used. Despite these facts, the sensor has several drawbacks, such as damage caused by heavy ion bombardment, which might cause specific interferences in the output signal [47].

Compared to other based assays, the piezoelectric transducer produced the lowest findings, with a LOD of 50 nM, utilizing SiO_2_ structure as the nanomaterial and aptamer as the detection target for dimethyl methylphosphonate. Furthermore, the preparation and experimental procedure were among the most cost-effective biosensors for the on-site detection of nerve agents and organophosphorus chemicals [47].

#### 3.2.6. SERS Transducer

SERS sensing systems use the magnetic and quantum principle to explain particle interactions and behavior, as well as the intensity of nanoparticles, which creates the Raman enhanced effect. SERS exhibits some parallels to optical transducer results while also exhibiting unique optical features for detection [36]. The SERS technique has the advantage of analyzing samples in their original condition and is unaffected by the sample detection mechanism or background. Because of these descriptions, SERS is becoming a key tool in various fields. However, various disadvantages of employing the SERS platform have been noted, such as a single NP displaying a weak signal and the unpredictable aggregation of nanoparticles resulting in unreproducible SERS signals, limiting their uses [36].

A biosensor based on the SERS transducer was developed to detect profenofos, acetamiprid, and carbendazim in rice and apple. A bimetallic nanomaterial Ag@AuNPs functionalized with particular aptamers for these analytes was used. The aptamer was utilized to adjust the distance between the Ag-Au nano-tetrahedron structure, which is implied in the SERS amplification effect, demonstrating that this approach has a high identification capability for food and environmental materials. This biosensor demonstrated high sensitivity, with LOD values of 2.1, 4.6, and 6.1 ng L^−1^ for profenofos, acetamiprid, and carbendazim, respectively [36].

### 3.3. Biosensors with Contaminant Analysis Devices

This section focuses on easy-to-build bio-contaminant detection devices that can help detect pesticides in food samples, providing nearly instant results compared to conventional analytical approaches [75] and can be measured and displayed using easily accessible equipment such as cell phone apps [44]. We might include the fluorescent aptamer-based lateral flow biosensor (apta-LFB), bidirectional lateral flow immunoassay (LFI) [44,75], and paper-based lateral flow biosensor (apta-LFB) [73].

For example, Cheng et al. used Pt nanoparticles anchored in two dimensions (2D) with Ni(OH)_2_ nanosheets (NSs) amplified by bidirectional LFI for the simultaneous detection of acetochlor and fenpropathrin in enriched maize, sorghum, soybean, apple, orange, peach, cabbage, broccoli, and tomato samples (see Figure 11A). Detected targets can be seen with the naked eye by observing the color changes of the test line after 13 min using a smartphone, and the Strip Scan analysis software can read the results in 10 min. Acetochlor and fenpropathrin had a LOD ranging from 6.3 × 10^2^ ng L^−1^ to 2.4 × 10^2^ ng L^−1^ [75]. Another device was built by Chen and colleagues which created a fluorescent apta-LFB; this system detects chlorpyrifos, diazinon, and malathion using the fluorescence of QDs nanobeads in conjunction with a smartphone spectrum reader, with LODs of 730 ng L^−1^, 6700 ng L^−1^, and 740 ng L^−1^, respectively, proving their great pesticide sensitivity [44]. Meanwhile, Liu et al. also created an LFB device to detect isocarbophos in Chinese cabbage, fresh tea leaves, and peach samples. The interaction of the aptamer with the pesticide is used in this LFB, and the amount of pesticide in the LFB’s test line (T) is directly proportional to the red color of the T zone, allowing the pesticide to be identified with the naked eye quickly [28] (Figure 11B).

Apilux et al. created a paper-based device based on fluorescence intensity. In order to simplify the multi-step assay and enhance the signal, the device was designed as a foldable sheet, including a detection zone with a buffer loading channel. The sample and buffer solutions can be placed on the paper in stages, and the measurement can be done with the naked eye. The device was used to detect organophosphorus and carbamates pesticides in lettuce, choy leaves, and rice grains, and the amount of analyte was evaluated by AChE inhibition. Following incubation, fluorescent photos of the detection zone are acquired with a digital camera, and the color change is quantified by measuring the intensity in the red channel. Pirimicarb, dichlorvos, and carbaryl had LODs of 5 × 10^4^, 1 × 10^4^, and 1 × 10^4^ ng L^−1^, respectively. Furthermore, the results were determined to be reasonable when compared to GC-MS/MS [73]. Chen et al. also received recovery values for biosensors that are identical to the GC method (83.4% to 110.7%); the results of this investigation are comparable to conventional analytical procedures (82.4% to 112.8%). Figure 11B shows how gold nanostars (AuNSs) were used in an apta-LFB-based lateral flow biosensor device capable of producing fluorescence in detection due to the connected QDs. The results of the analytical parameters show the device’s accuracy and viability in different detection matrices [44]. In another example, Cheng et al. used gas chromatography to examine the outcomes of non-enriched materials and found “not detected” results in both investigations [75]. Another advantage is the timeliness with which the detection process is completed. According to Liu et al., the overall detection time was around 20 min, the sample pretreatment time was about 15 min, and the detection results were visible in 1 min [28]. As a result, these devices are portrayed as inexpensive, quick, and simple alternatives that do not require highly trained professionals to conduct the analyses. Figure 11 depicts some of the studies aforementioned in this section.

### 3.4. Highlights and Futures Perspectives

It can be observed that some articles demonstrated the evolution of novel nanomaterial and transducing techniques while others demonstrated reliable, robust, and simple to use transducing materials. The biosensor developed by Chen Ge et al., designated as bio-immunoassay based on the catalysis of bimetallic nanoenzymes (Au@Pt), is not widely employed in the detection of the pesticide par-athion. Three probes were utilized for this: (i) AuNPs were treated with oligonucleotides and monoclonal antibodies (mAbs), (ii) Magnetic nanoparticles (MNPs) were coated with ovalbumin haptens (OVA)-parathion, and (iii) bimetallic nanoenzymes (Au@Pt) nanoparticles were functionalized with ssDNA. A magnetic field was used to separate the Au@Pt nanoenzyme from the complexes such that it could catalyze 3,3′,5,5′-tetramethylbenzidine (TMB). To validate the method’s feasibility, a bio-barcode immunoassay based on bimetallic nanoenzymes was performed on rice, pear, apple, and cabbage samples. The immunoassay based on Au@Pt nanoenzyme catalysis showed a linear response from 0.01 to 40 μg·kg^−1^ with a LOD of 2.13 × 10^−3^ μg·kg^−1^ and was found to correlate with the LC-MS/MS method [46].

Multi-residue pesticide biosensors capable of concurrently detecting more than one pesticide are the subject of exemplary research. In such cases, Zhao et al. developed an electrochemical biosensor using nanogold/mercaptomethamidophos to detect 12 pesticides of the organophosphate family (Au-S). The mercaptomethamidophos functionalized in AuNPs binds strongly to AChE, allowing pesticides to be detected by the indirect competitive technique. The study showed excellent electrochemical properties with a wide linear range of 0.1~1500 ng·mL^−1^ and LOD 0.019~0.077 ng·mL^−1^. This biosensor was applied to food samples of cabbage and apple, detecting trichlorfon and dichlorvos pesticides [24]. This biosensor demonstrated has a high potential for use in environmental detection and food safety. Figure 12 depicts the author’s description of the production of the electrochemical biosensor, voltammograms of the 11 distinct pesticides, calibration curve, and analysis time.

Sensors based on nanozyme, which combine the properties of a nanomaterial and a bioenzyme, have been developed to overcome the challenges associated while using natural enzymes. However, as it is based primarily on a colorimetric approach, it is prone to severe color interference induced by the nanozymes and samples themselves. Several efforts have been attempted in recent years to overcome color interference. Recently, a promising electrochemical sensor based on nanozymes with manganese dioxide nanosheets (MnNS) for the detection of Ops was proposed by Wu et al. [103]. According to the authors, the MnNS eliminated color interference and allowed the detection of paraoxon in the pak choi sample with high precision and good accuracy, with a LOD of 0.025 ng mL^−1^.

The studies included for this systematic review presented the most diverse types of nanomaterials, either pure or hybrid, that can be employed in pesticide biosensors. The pure forms are already established in the literature; however, a combination, such as the hybrid form, integrates the properties of both materials, yielding better analytical results. Future development will increasingly focus on combining characteristics to create more sensitive and selective biosensors. Another possibility for future research is to use different biorecognition materials such as aptamer, antibodies, DNA, and nanoenzymes. Because AChE is used in most research, it is not as selective as an oligonucleotide synthesized, particularly for the target analytes. Finally, future prospects include the development of multi-residue detection biosensors that are portable and may be used in the field, overcoming the constraints of traditional detection methods.

### 3.5. Efficiency of Biosensors in Legislation: MRL, ARfD and LD_50_

The application of field-developed biosensors depends directly on the obtained LODs, assessing whether they can detect low concentrations at trace levels of pesticide. Pesticide MRLs should always be considered in reliability studies of pesticide techniques [107,108]. As a result, the low readings were compared to the MLRs established by the Codex Alimentarius [109] and the United States Environmental Protection Agency (USEPA) [110]. All LOD values achieved in each investigation are lower than the MRL for each pesticide in the evaluated foods, which proves a strong sensitivity, especially in experiments that used AChE-based electrochemical biosensors. The LOD values obtained for each biosensor and the MRL values of the pesticides can be consulted in more detail in the Appendix A. In addition, to verify whether or not the biosensors met the requirements to detect concentrations of risk to human health, the LOD values were compared with toxicological data from Acute Reference Doses (ARfD), Acceptable daily intake (ADI), and Median Lethal Dose (LD_50_) that are established by the USEPA [110] (Table 4). All biosensors evaluated in this study showed LOD values much lower than the pesticide concentration that can cause harm to human health. Therefore, these devices can be considered efficient in detecting small concentrations and consequently helping to control acceptable levels of pesticides established by regulatory agencies.

## 4. Conclusions

Biosensors could be a more cost-effective, quicker, and more resilient alternative to standard chromatographic methods for detecting pesticides in foods. According to the findings, detection may differ based on the nanomaterial, biorecognition material, analyte of interest, and transducer employed to generate response signals. The most commonly utilized transducers were electrochemical ones, which stand out for their great sensitivity and ease of use. Furthermore, colorimetric transducers stand out due to their high sensitivity and ability to detect with the naked eye, which is their main advantage. Fluorescence, SERS, and piezoelectric transducers were also used and demonstrated good sensitivity; however, they were used less frequently.

Nanomaterials are being employed as an option to improve the sensitivity of biosensors. AuNPs, AgNPs, CNTs, and nanohybrids are the most often used nanomaterials. The employment of magnetic nanoparticles as sample preparation in a magnetic detecting system was a highlight. Not only do nanomaterials aid/influence the analytical signals of a biosensor, but biological recognition materials also play an important role in these devices, as they are responsible for pesticide chemical interaction. According to the investigations, the most commonly employed biological materials among these biomaterials were AChE and aptamers.

Food is one of the main sources of pesticide contamination; fruits, vegetables, and grains are some examples of food matrices widely recorded. In this regard, the biosensors produced in the selected experiments helped detect pesticides in food, with LODs attained compared to the Codex Alimentarius MRLs. As a result, biosensors can be used to identify pesticides in food because they can detect values lower than those needed by food standards.

## Figures and Tables

**Figure 1 biosensors-12-00572-f001:**
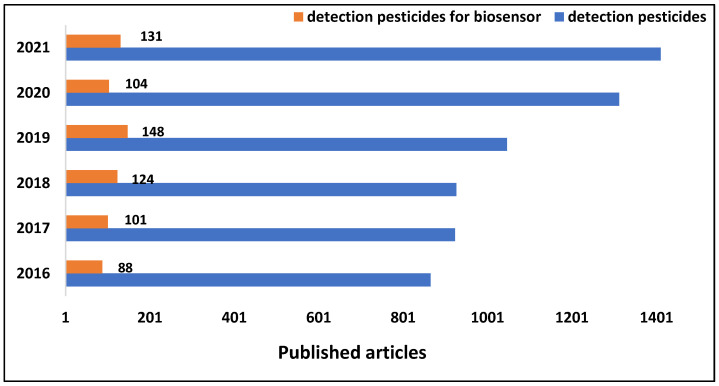
Published articles in the database Web of Science to detect pesticides in the years 2016–2022 as of May 25. Foam search strings: biosensor AND food AND (agrochemical OR pesticides), restricted only to original search articles.

**Figure 2 biosensors-12-00572-f002:**
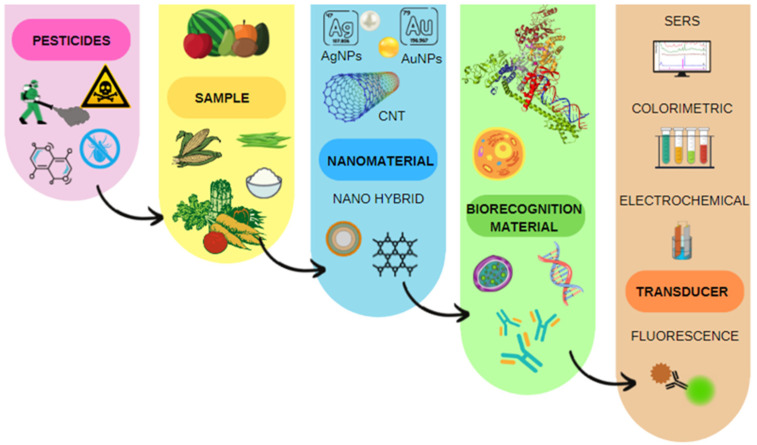
Schematic of the steps that are necessary to build a biosensor.

**Figure 3 biosensors-12-00572-f003:**
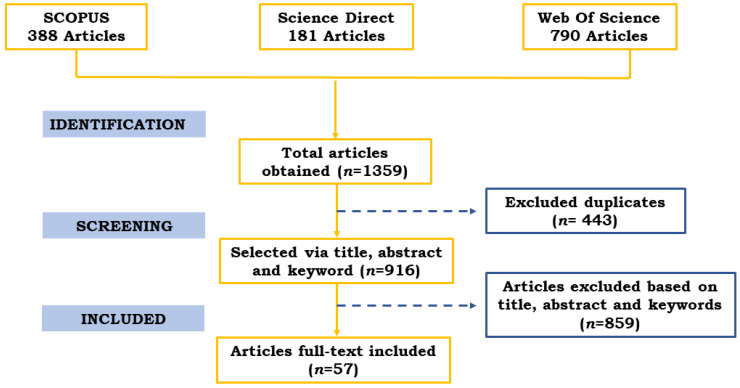
PRISMA flow diagram displaying systematic search results, indicating the total number of articles obtained from the databases (*n* = 1359) and those selected according to the inclusion criteria (*n* = 57).

**Figure 4 biosensors-12-00572-f004:**
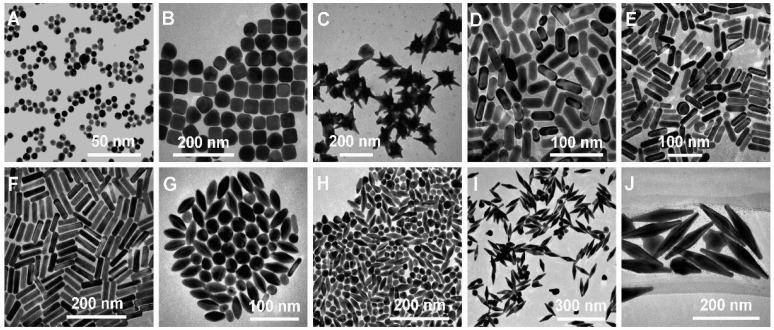
Transmission Electron Microscopy (TEM) images of different AuNPs of different sizes and shapes. (**A**) Nanospheres, (**B**) Nanocubes, (**C**) Nanobranches, (**D**) Nanorods, (**E**) Nanorods, (**F**) Nanorods, (**G**) Nanobipyramids, (**H**) Nanobipyramids, (**I**) Nanobipyramids, and (**J**) Nanobipyramids. Reproduced with permission from Chen et al. [86]. Copyright 2008, American Chemical Society.

**Figure 5 biosensors-12-00572-f005:**
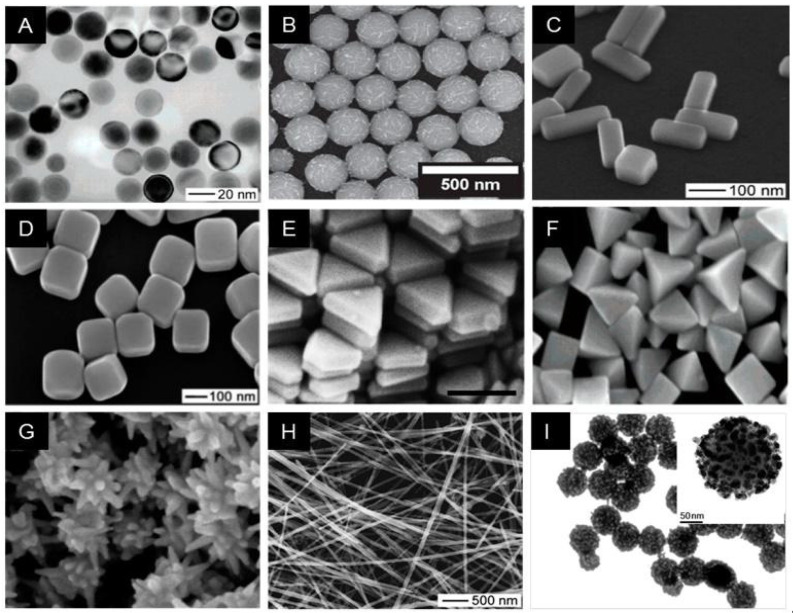
Transmission Electron Microscopy (TEM) images of different AuNPs of different sizes and shapes. (**A**) Silver nanosphere, (**B**) Silver necklaces, (**C**) Silver nanobars, (**D**) Silver nanocubes, (**E**) Silver nanoprism, (**F**) Silver bipyramids, (**G**) Silver nanostar, (**H**) Silver nanowire, and (**I**) Silver nanoparticle embedded silica particle. Reproduced with permission from Lee et al. [87]. Copyright 2019, MDPI.

**Figure 6 biosensors-12-00572-f006:**
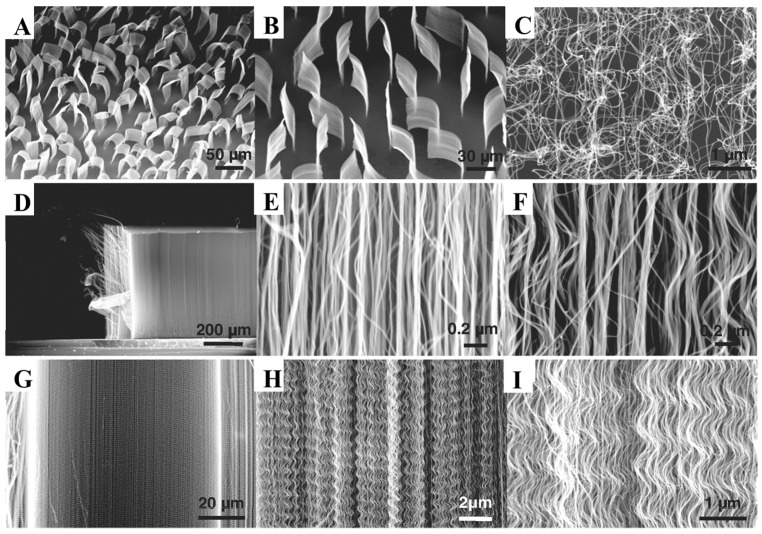
SEM images of different carbon nanotubes (CNT) at different magnifications in thin sheets (**A**–**D**). Additionally super-aligned, Straight, and waved MWNTs array (**E**–**I**). Reproduced with permission from Zhang et al. [90] Copyright 2009, Elsevier.

**Figure 7 biosensors-12-00572-f007:**
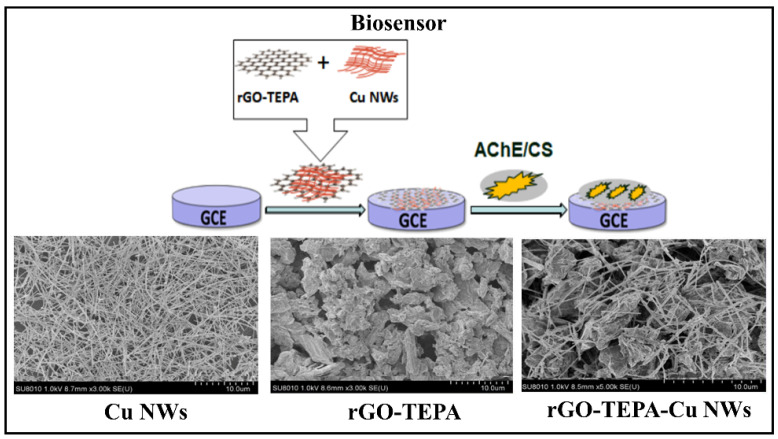
Fabrication of AChE-Cs/rGO-TEPA-Cu NWs/GCE biosensor and SEM images CuNWs, rGO-TEPA and rGO-TEPA-CuNWs. Reproduced with permission from Li et al. [70] Copyright 2020, ESG (open access journal).

**Figure 8 biosensors-12-00572-f008:**
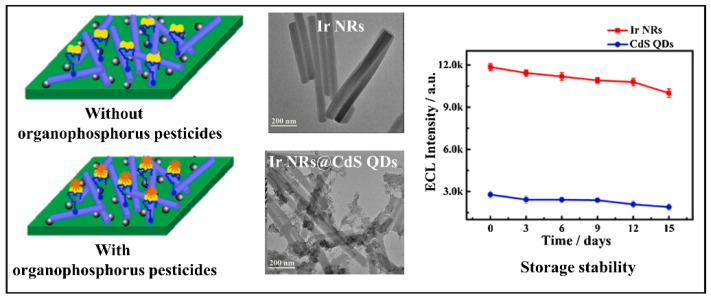
The schematic description of the actuation of the biosensor in OPs presence and TEM images of Ir NRs, and Ir NRs@CdS QDs. Reproduced with permission from Yang et al. [76]. Copyright 2021, Elsevier.

**Figure 9 biosensors-12-00572-f009:**
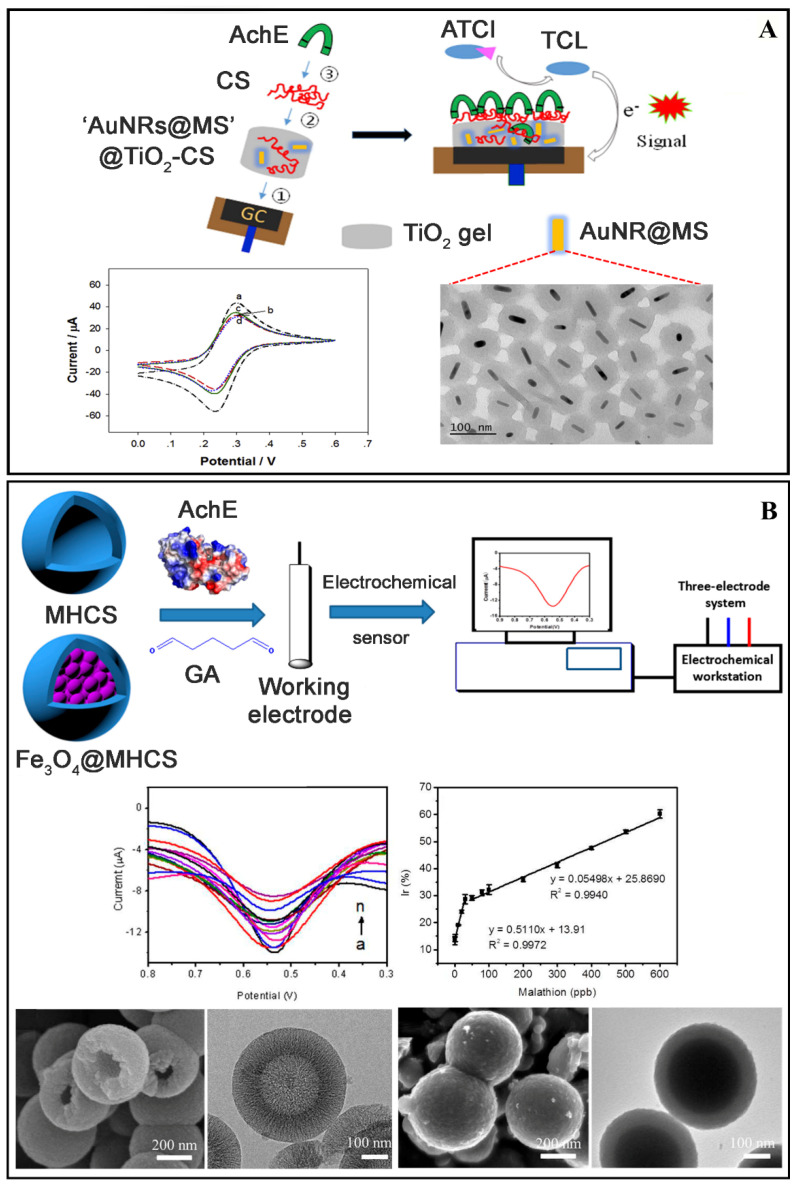
(**A**) Schematic illustration of the fabrication steps and the structure of the AuNRs@MS nanoparticles doped AChE biosensor and its working mechanism to acetylthiocholine evidenced the formed nanostructure and the type of electrochemical detection. Reproduced with permission from Cui et al. [38]. Copyright 2019, Elsevier. (**B**) Schematic illustration of the preparation of acetylcholinesterase (AChE)/carbon core-shell structures (Fe_3_O_4_@MHCS)/GCE electrochemical sensors. Evidenced images were obtained by Scanning electron microscopy (SEM), electrochemical detection, and analytical linearity parameter. Reproduced with permission from Luo et al. [42]. Copyright 2018, MDPI.

**Figure 10 biosensors-12-00572-f010:**
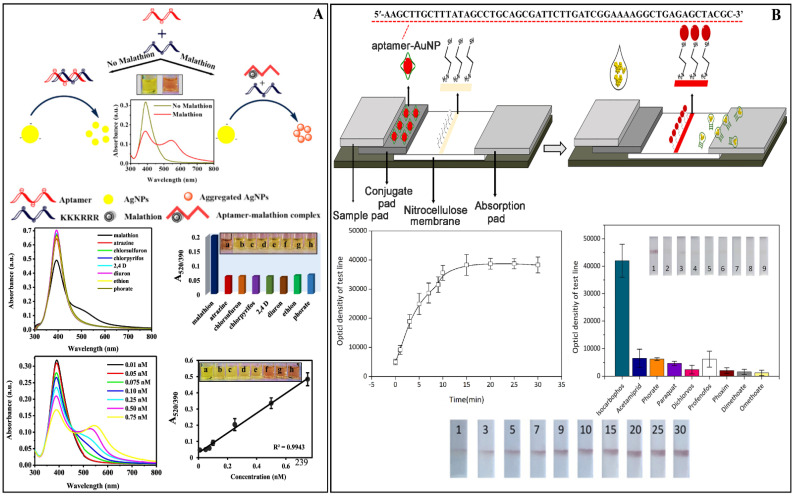
(**A**) Schematic representation of the detection strategy for malathion by employing optical properties of silver nanoparticles. Evaluation of the selectivity of the biosensor for the detection of malathion evaluated with other pesticides and sensitivity through the analytical curves constructed. Reprinted with permission from Bala et al. [34]. Copyright 2018, Elsevier. (**B**) Isocarbophos detection on lateral flow biosensor using AuNP-aptamers probe, indicating the time required for the detection of 30 min and selectivity test in the presence of other pesticides. Reprinted with permission from Liu et al. [28]. Copyright 2021, Elsevier.

**Figure 11 biosensors-12-00572-f011:**
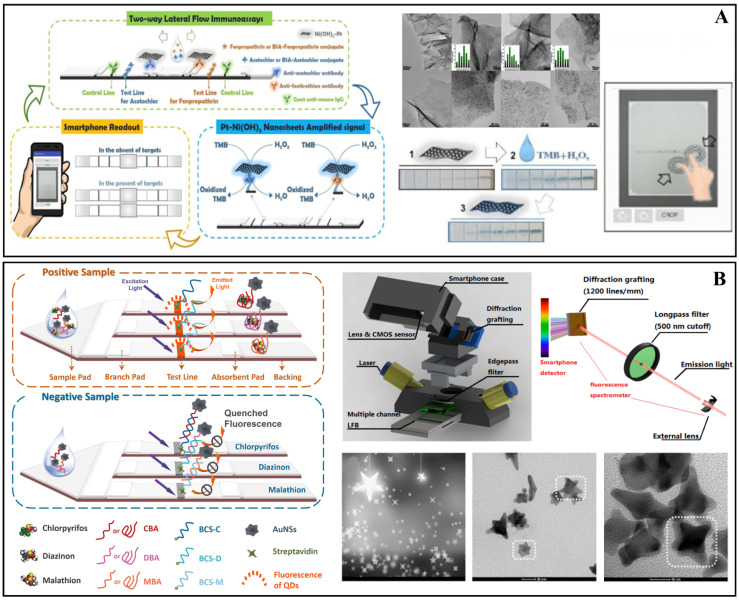
(**A**) Portable system for rapid assays of herbicide and insecticide residues, featuring schematic diagram of two-way LFI for detection of acetochlor and fenpropathrin. Transmission electron microscopy (TEM) images of as-prepared Ni(OH)_2_ nanosheets, and LFI images during the reaction. Reprinted with permission from Cheng et al. [75]. Copyright 2019, Elsevier. (**B**) Development of a fluorescent aptamer-based lateral flow biosensor (apta-LFB) with design: positive and negative samples for the pesticides chlorpyrifos, diazinon, and malathion. Reprinted with permission from Chen et al. [44]. Copyright 2018, Elsevier.

**Figure 12 biosensors-12-00572-f012:**
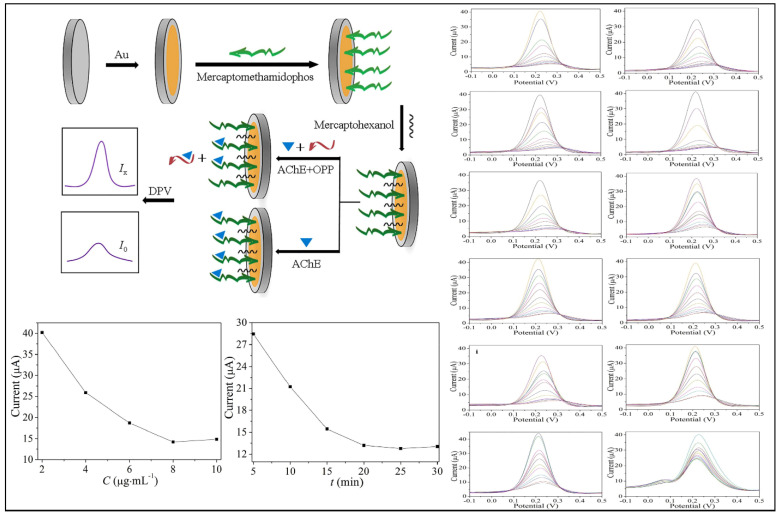
Detection illustration of nanogold/mercaptomethamidophos electrochemical biosensor, AChE concentration added, incubation time, and differential pulse voltammetry of the 12 pesticides detections. Reprinted with permission from Zhao et al. [24]. Copyright 2021, Elsevier.

**Table 1 biosensors-12-00572-t001:** Inclusion and exclusion criteria used for the selection of articles.

Inclusion Criteria	Exclusion Criteria
English language	Non-English language articles
Original research articles	Thesis, review articles, and short communications
Use of biosensors for pesticide detection	Use of biosensors to detect non-pesticide
Biosensor application in food matrices	Biosensors not applied in food matrices
Articles published from 2016 to 2021	Articles published outside of this timeline

**Table 2 biosensors-12-00572-t002:** Description of biosensors focusing on the type of nanomaterial used.

Nanomaterial	Biorecognition Material	LOD	Pesticide or Pesticide Class	Food Matrix	Ref.
AuNPs	AChE	Organophosphorus: 19–77 ng L^−1^ Methomyl: 81 ng L^−1^	11 Organophosphorus pesticides and Methomyl	Apple and Cabbage	[24]
AuNPs	AChE	1.0 nM	Carbamate	Fruit	[25]
AuNPs	Aptamer	36 ng L^−1^	Chlorpyrifos	Apple and Pak choi	[26]
AuNPs	Antibody	70 × 10^−3^ ng L^−1^	Chlorpyrifos	Chinese cabbage and Lettuce	[27]
AuNPs	Aptamer	2.48 × 10^3^ ng L^−1^	Isocarbophos	Cabbage, Peach and Tea	[28]
PEDOT-MWCNTs	Antibody	1 × 10^−6^ nM	Malathion	Lettuce	[29]
AuNPs CP-MOF-Fc	Aptamer	17.18 ng L^−1^	Malathion	Cucumber and Long bean	[30]
AgNPs GQDs	AChE	17 × 10^3^ ng L^−1^	Paraoxon	Apple and Carrot	[31]
AgNPs	G-DNA	34 ng L^−1^	Organophosphorus	Apple	[32]
AgNPs	AChE	4 × 10^3^ ng L^−1^	Paraoxon	Chives and Cabbage	[33]
AgNPs	Aptamer	5 × 10^−4^ nM	Malathion	Apple	[34]
AgNWs	BChE	212 nM	Paraoxon	Milk	[35]
Ag@AuNPs	Aptamers	Profenofos: 2.1 ng L^−1^; Acetamiprid: 4.6 ng L^−1^ Carbendazim: 6.1 ng L^−1^	Profenophos, Carbendazim and Acetamiprid	Rice and Apple	[36]
Au–Ag NC	BSA	8.2 nmol L^−1^	Methyl parathion	Apple, Cabbage, Spinach and Lettuce	[37]
AuNRs and MS SiO_2_	AChE	Fenthion: 1.3 nM Dichlovos: 5.3 nM	Fenthion and Dichlorvos	Cabbage juice	[38]
AuNPs and UCNPs	ABA	0.36 nM	Acetamiprid	Celery leaves and Chinese green tea	[39]
AuNPs and VNSWCNTs	AChE	Methyl parathion: 3.04 × 10^−3^ ng L^−1^ Malathion: 1.96 × 10^−3^ ng L^−1^ Chlorpyrifos: 2.06 × 10^−3^ ng L^−1^	Methyl parathion, Malathion and Chlorpyrifos	Cabbage juice	[40]
Au-Ag NC	AChE	2.40 × 10^−3^ nM	Ethyl parathion	Orange and Apple juice	[41]
MHCS and Fe_3_O_4_@MHCS	AChE	MHCS: 14.8 ng L^−1^ Fe_3_O_4_@MHCS: 18.2 ng L^−1^	Malathion	Pear	[42]
PtPd@NCS	AChE	8.6 × 10^−6^–7.1 × 10^−5^ nM	Malathion, Chlorpyrifos and Methyl parathion	Potato and Corn grans	[43]
QDs-AuNSs	Antibodies	Chlorpyrifos: 730 ng L^−1^ Diazinon: 6.7 × 10^3^ ng L^−1^ Malathion: 740 ng L^−1^	Chlorpyrifos, Malathion and Diazinon	Maize, Long bean, Cauliflower, Eggplant, Oyster mushroom, Shiitake mushroom, Apple, Orange, Tomato, Blueberry, Spinach, Lettuce and Cabbage	[44]
PtNPs AuNPs and MNPs	mAbs ssDNA C-ssDNA	2 ng L^−1^	Parathion	Pear, Cabbage and Rice	[45]
Au@PtNPs MNPs	ssDNAs and mAbs	2.13 ng kg^−1^	Parathion	Rice, Pear, Apple and Cabbage	[46]
SiO_2_ and Cr/Au modified layer	Aptamer	50 nM	Dimethyl-methylphosphonate	Apple juice	[47]
PDA-AuNPs	Aptamer	5 × 10^−1^ ng L^−1^	Malathion	Cauliflower and Cabbage	[48]
AuNPs	AChE	1.4 × 10^3^ ng L^−1^	Paraoxon	Vegetable (not specified)	[49]
MWCNTs	AChE	50 ng L^−1^.	Chlorpyrifos	Cabbage, Rape and Lettuce	[50]
MWCNTs	AChE	1 × 10^−6^ ng L^−1^	Malathion	Lettuce	[51]
MWCNTs	ds-DNA	0.3 nmol L^−1^	Diazinon	Lettuce and Tomato juice	[52]
MWCNT	AChE	0.1 nM	Paraoxon	Potato	[53]
MWCNTs	AChE	4 × 10^−3^ nM	Organophosphate	Spinach and Cabbage	[54]
f-MWCNTs	AChE	1 ng L^−1^	Chlorpyrifos-methyl	Lettuce	[55]
ZnS:Mn-QDs and MWCNTs	Aptamer	0.7 nM	Acetamiprid	Cabbage leaves	[56]
CNT	M-Cell	3 × 10^−6^ nmol L^−1^	Paraoxon	Spinach juice	[57]
PB-SWCNTs	AChE	Malathion: 3.11 × 10^−4^ ng L^−1^ Metyl parathion: 1.88 × 10^−4^ ng L^−1^	Malathiona and Methyl parathion	Chinese cabbage	[58]
Fe_3_O_4_ and graphene	AChE	20 ng L^−1^	Chlorpyrifos	Cabbage and Spinach	[59]
TiO_2_ NP	AChE	0.23 nM	Dichlorvos	Cabbage juice	[60]
TiO_2_NP	Nanoenzymes	Methyl paraoxon: 240 nM Methyl parathion: 260 nM Ethyl paraoxon: 220 nM	Organophosphorus	Lettuce	[61]
Film titanium with AuNP	Aptamer	1.3 × 10^3^ ng L^−1^	Profenofos	Chinese chives	[62]
SBA-15	AChE	Monocrotophos: 2510 ng L^−1^ Dimethoate: 1500 ng L^−1^	Monocrotophs and Dimethoate	Soft drinks	[63]
WO_3_/g-C_3_N_4_	Tc-AChE	3.6 nM	Phosmet	Wheat flour	[8]
CS-PVA	AChE	0.2 nM	Pirimiphosmethyl	Olive oil	[64]
CdTe-QD	AChE and CHOx	Paraoxon: 1.62 × 10^−6^ nM Dichlorvos: 7.53 × 10^−5^ nM Malathion: 0.23 nM Triazophos: 1.06 × 10^−2^ nM	Paraoxon, Dichlorvos, Malathion and Triazophos	Apple and Tomato juice	[65]
CHIT-IO	Biotinylated DNA	1 ng L^−1^	Malathion	Lettuce leaves	[66]
rGO/AuNPs	AChE	Malathion: 2.78 × 10^−2^ ng L^−1^ Methyl parathion: 2.17 × 10^−2^ ng L^−1^	Malathion and Methyl parathion	Chinese cabbage	[67]
rGO	AChE	1.9 nmol L^−1^	Carbamate	Tomatoes	[68]
rGO	Aptamer	7.12 × 10^−5^ nM	Acetamiprid	Tea	[69]
rGO-TEPA-CuNW	AChE	3.9 × 10^2^ ng L^−1^	Malathion	Cabbage and Carrot	[70]
CS@TiO_2_-CS/rGO	AChE	29 nM	Dichlorvos	Cabbage juice	[71]
ZIF-8	AChE	1.70 × 10^3^ ng/L	Paraoxon	Apple and Eggplant	[31]
MPtPdN	AChE	1.7 × 10^−3^ nM	Organophosphate	Cabbage and Cucumber	[72]
CdTe-QD	AChE	Pirimicarb: 5 × 10^4^ ng L^−1^ Dichlorvos: 1 × 10^4^ ng L^−1^ Carbaryl: 1 × 10^4^ ng L^−1^	Organophosphorus and Carbamate	Lettuce, Choy and Rice	[73]
Nanocarriers (Proline- UIO-66)	Candida Rugosa Lipase	26 nM	Nitrofen	Apricot	[74]
Pt–Ni(OH)_2_ and nanosheets	Antibodies and Nitrocellulose membrane	Acetochlor: 6.3 × 10^2^ ng L^−1^ Fenpropathrin: 2.4 × 10^2^ ng L^−1^	Acetochlor and Fenpropathrin	Corn, Sorghum, Soybean, Apple, Orange, Peach, Cabbage, Broccoli, Tomato and Drinking water	[75]
Ir NRs@CdS QDs	AChE-ChOx biocomposite	1.67 × 10^−3^ nM	Organophosphorus	Pakchoi, Cabbage and Lettuce	[76]
UCNPs	Aptamer	50 ng⋅L^−1^	Carbendazim	Apple, Cucumber and Matcha powder	[77]
CuO NFs and c-SWCNTs	Oligonucleotides	70 ng L^−1^	Chlorpyrifos	Apple and Cabbage	[78]

**Legend:** (ABA) Acetamiprid-binding aptamer; (AChE) Acetylcholinesterase; (AgNPs) Silver nanoparticles; (AgNWs) Silver nanowires; (AuNPs) Gold nanoparticles; (AuNRs) Gold nanorods; (AuNSs) Gold nanostars; (BChE) Butyrylcholinesterase; (BSA) Bovine serum albumin; (C-ssDNA) Complementary ssDNA; (c-SWCNTs) Carboxyl-functionalized single-walled carbon nanotubes; (CNT) Carbon nanotubes; (CS) Chitosan; (CHIT-IO) Chitosan-iron oxide; (CHOx) Choline oxidase; (CP) Complementary probe; (Cu NWs) Copper nanowires; (CuO NFs) Nanoflowers; (ds-DNA) Double-stranded DNA; (f-MWCNTs) Functionalized multi-walled carbon nanotubes; (Fc) Ferrocene; (g-C_3_N_4_) Graphitic carbon nitride; (G-DNA) Guanine-rich DNA; (GQDs) Graphene quantum dots; (GR) Graphene; (Ir) Iridium; (M-Cell) Mineralized cell; (mAbs) Monoclonal antibodies; (MHCS) Mesoporous hollow carbon spheres; (MNPs) Magnetic nanoparticles; (MOF) metal organic framework; (MPH) methyl parathion hydrolase; (MPtPdN) Mesoporous bimetallic PtPd nanoflowers; (MS) Mesoporous; (MWCNTs) Multi-walled carbon nanotubes; (NC) Nanocluster; (NCDs) N-doped carbon dots; (NCS) N-doped carbon shells; (NPs) Nanoparticles; (NRs) Nanorods; (NSs) Nanosheets; (PB) Prussian blue; (PDA) Polydopamine; (PEDOT) Poly(3,4-ethylenedioxythiophene); (PVA) Poly (vinyl alcohol); (QDs) Quantum dots; (rGO) Reduced graphene oxide; (SBA) Santa Barbara Amorphous; (SERS) Surface Enhanced Raman Spectroscopy; (ssDNA) Single-stranded DNA; (SWCNTs) Single-wall carbon nanotubes; (TC) Tribolium castaneum; (TEPA) Tetraethylenepentamine; (UCNPs) Upconversion nanoparticles; (VNSWCNTs) Vertical nitrogen-doped single-walled carbon nanotubes; Quantum Dots of Cadmium Sulfide combined with Iridium nanorods (Ir NRs@CdS QDs).

**Table 3 biosensors-12-00572-t003:** Transducer-based biosensors for the detection of pesticides in food.

Biosensor-Based	Biorecognition Material	Pesticide or Pesticide Class	Transducer Type	LOD	RSD(%)	Ref.
DNA	Aptamer	Malathion	Colorimetric	5 × 10^−4^ nM	2.98	[34]
DNA	C-ssDNA	Parathion	Colorimetric	2 ng L^−1^	Pear: 5.19 Cabagge: 9.81 Rice: 15.75	[45]
Enzyme	Nanozyme	Parathion	Colorimetric	2.13 ng kg^−1^	Rice: 5.59 Pear: 6.09 Apple: 10.18 Cabbage: 10.87	[46]
DNA	Aptamer	Isocarbophos	Colorimetric	2.48 × 10 ^3^ ng L^−1^	2.37–7.13	[28]
Antibodies	Antibodies	Acetochlor and Fenpropathrin	Colorimetric	Acetochlor: 6.3 × 10^2^ ng L^−1^ Fenpropathrin: 2.4 × 10^2^ ng L^−1^	3.30	[75]
Enzyme	BChE	Paraoxon	Electrochemical (Amperometric)	212 nM	-	[35]
Enzyme	AChE	Paraoxon	Electrochemical (CV)	4 × 10^3^ ng L^−1^	Chinese chives: 2.39 Cabbage: 5.86	[33]
Antibodies	BSA	Methyl parathion	Electrochemical (CV)	8.2 nmol L^−1^	4.7	[37]
DNA	Oligonucleotides	Chlorpyrifos	Electrochemical (CV)	70 ng L^−1^	Apple: 2.52 Celery cabbage: 2.25	[78]
Enzyme	AChE	Chlorpyrifos	Electrochemical	20 ng L^−1^	Cabbage: 3.86 Spinach: 2.46	[59]
Enzyme	AChE	Chlorpyrifos	Electrochemical (CV)	50 ng L^−1^	Cabbage: 4.35 Rape: 2.57 Lettuce: 3.17	[50]
Enzyme	AChE	Dichlorvos	Electrochemical (CV and DPV)	0.23 nM	7.3	[60]
Enzyme	AChE	Malathion	Electrochemical (DPV)	1 × 10^−6^ nM	-	[51]
Enzyme	AChE	Malathion and Methyl parathion	Electrochemical (CV)	Malathion: 3.11 × 10^−4^ ng L^−1^ Methyl parathion: 1.88 × 10^−4^ ng L^−1^	4.59	[58]
Enzyme	AChE	Monocrotophos and Dimethoate	Electrochemical (CV)	Monocrotophos 2.51 × 10^3^ ng L^−1^ Dimethoate 1.50 × 10^3^ ng L^−1^	Monocrotophos: 1.05 Dimethoate: 0.95	[63]
Enzyme	AChE	Fenthion	Electrochemical (CV and EIS)	1.3 nM	11.5	[38]
Enzyme	AChE	Chlorpyrifos-methyl	Electrochemical (DPV)	1 ng L^−1^	-	[55]
DNA	ds-DNA	Diazinon	Electrochemical (EIS)	0.3 nmol L^−1^	-	[52]
Enzyme	Tc-AChE	Phosmet	Electrochemical (CV and EIS)	3.6 nM	2.5	[8]
Enzyme	AChE	Malathion	Electrochemical (CV and EIS)	3.9 × 10^2^ ng L^−1^	2.3	[70]
Enzyme	AChE	Pirimiphos methyl	Electrochemical (Amperometric)	0.2 nM	-	[64]
Enzyme	AChE and CHOx	Malathion	Electrochemical (DPV)	1 ng L^−1^	-	[66]
Enzyme	AChE	11 Organophosphorus pesticides and Methomyl	Electrochemical (DPV and EIS)	Organophosphorus: 19–77 ng L^−1^ Methomyl: 81 ng L^−1^	Trichlorfon: 1.80–8.63 Dichlorvos: 3.21–9.20	[24]
Enzyme	Nanoenzyme	Organophosphorus	Electrochemical (DPV)	Methyl paraoxon: 240 nM Methyl parathion: 260 nM Ethyl paraoxon: 220 nM	Methyl paraoxon: 3.41 Methyl parathion: 2.41 Ethyl paraoxon: 2.56	[61]
DNA	Aptamer	Chlopyrifos	Electrochemical (CV)	36 ng L^−1^	2.57–7.08	[26]
Enzyme	AChE	Organophosphorus	Electrochemical (CV)	Malathion: 2.78 × 10^−2^ ng L^−1^ Methyl parathion: 2.17 × 10^−2^ ng L^−1^	4.07	[67]
Enzyme	AChE	Paraoxon	Electrochemical (CV)	0.1 nM and 500 nM	-	[53]
Enzyme	AChE	Paraoxon	Electrochemical (DPV)	1.4 × 10^3^ ng L^−1^	4.68	[49]
Enzyme	AChE	Malathion	Electrochemical (CV and EIS)	3.9 × 10^2^ ng L^−1^	2.30	[70]
Enzyme	AChE	Organophosphorus	Electrochemical (CV and EIS)	14.8 ng L^−1^–18.2 ng L^−1^	5.6–7.1	[42]
Enzyme	AChE	Carbaryl	Electrochemical (CV)	1.0 nM	5.32	[25]
Enzyme	AChE	Malathion, Chlorpyrifos and Methyl parathion	Electrochemical (DPV)	8.6 × 10^−6^–7.1 × 10^−5^ nM	3.31–5.24	[43]
Enzyme	AChE	Methyl parathion, Malathion and Chlorpyrifos	Electrochemical (CV, DPV and EIS)	Methyl parathion: 3.04 × 10^−3^ ng L^−1^ Malathion: 1.96 × 10^−3^ ng L^−1^ Chlorpyrifos: 2.06 × 10^−3^ ng L^−1^	3.74	[40]
Enzyme	AChE	Carbaryl	Electrochemical (CV and EIS)	1.9 nmol L^−1^	-	[68]
Enzyme	Candida Rugosa Lipase	Nitrofen	Electrochemical (DPV)	26 nM	1.75–4.12	[74]
Enzyme	AChE-ChOx	Organophosphorus	Electrochemical (CV)	1.67 × 10^−3^ nM	5	[76]
Cell	M-Cell	Paraoxon	Electrochemical (DPV)	3 × 10^−6^ nmol L^−1^	3.5	[57]
Antibodies	Antibody	Chlorpyrifos	Electrochemical (EIS)	70 × 10^−3^ ng L^−1^	2.6	[27]
Enzyme	AChE	Dichlorvos	Electrochemical (CV and EIS)	29 nM	1.44	[71]
DNA	Aptamer	Acetamiprid	Electrochemical (CV and EIS)	7.12 × 10^−5^ nM	5.9	[69]
Enzyme	AChE	Paraoxon	Electrochemical (CV and EIS)	4 × 10^−3^ nM	Spinach: 10.2–96 Cabbage: 3.4–4.1	[54]
Enzyme	AChE	Organophosphorus	Electrochemical (DPV)	1.73 × 10^−3^ nM	3.84 and 5.91	[72]
DNA	Aptamer	Malathion	Eletrochemical (DPV)	5 × 10^−1^ ng L ^−1^	1.04–6.14	[48]
Enzyme	AChE	Paraoxon	Electrochemical (DPV)	1.7 × 10 ^3^ ng L^−1^	Apple: 3.2–3.7 Eggplant 5.0–4.6	[31]
DNA	Aptamer	Malathion	Electrochemical (DPV)	17.18 ng L^−1^	Cucumber: 1.17–1.33 Beans: 0.52–5.90	[30]
Antibodies	Antibodies	Malathion	Electrochemical (DPV)	1 × 10^−6^ nM	1.15–3.21	[29]
DNA	Aptamer	Cabendazim	Fluorescence	50 ng L^−1^	Apple: 2.02–4.39 Cucumber: 2.90–4.30 Matcha powder: 1.87–3.51	[77]
DNA	G-DNA	Organophosphorus	Fluorescence	34 ng L^−1^	Apple: < 2.8	[61]
DNA	Aptamer	Chlorpyrifos, Diazinon and Malathion	Fluorescence	Chlorpyrifos: 730 ng L^−1^ Diazinon: 6.7 × 10^3^ ng L^−1^ Malathion: 740 ng L^−1^	-	[44]
Enzyme	AChE and CHOx	Paraoxon, Dichlorvos, Malathion and Triazophos	Fluorescence	1.62 × 10 ^−6^–0.23 nM	2.23–7.19	[65]
DNA	ABA	Acetamiprid	Fluorescence	0.36 nM	<4.54	[39]
Enzyme	AChE	Pirimicarb, Dichlorvos and Carbaryl	Fluorescence	Pirimicarb: 5 × 10^4^ ng L^−1^ Dichlorvos: 1 × 10^4^ ng L^−1^ Carbaryl: 1 × 10 ^4^ ng L^−1^	-	[73]
Enzyme	AChE	Ethylparathion	Fluorescence	2.40 × 10^−3^ nM	-	[41]
DNA	Aptamer	Acetamiprid	Fluorescence	0.7 nM	Cabbage leaves: 1.0–2.1	[56]
DNA	Aptamer	Profenofos	Microcantilever	1.3 × 10^3^ ng L^−1^	-	[62]
DNA	Aptamer	Dimethyl methylphosphonate	Piezoelectric	50 nM	-	[47]
DNA	Aptamer	Profenofos, Acetamiprid and Carbendazim	SERS	Profenofos: 2.1 ng L^−1^ Acetamiprid: 4.6 ng L^−1^ Carbendazin: 6.1 ng L^−1^	-	[36]

**Legend:** (-) not reported (ABA) Acetamiprid-binding aptamer; (AChE) Acetylcholinesterase; (BChE) Butyrylcholinesterase; (BSA) Bovine serum albumin; (C-ssDNA) Complementary ssDNA; (CHOx) Choline oxidase; (ds-DNA) Double-stranded DNA; (G-DNA) Guanine-rich DNA; (M-Cell) Mineralized cell; (SERS) Surface Enhanced Raman Spectroscopy; (Tc) Tribolium castaneum; voltametria de pulso diferencial (DPV); espectroscopia de impedância eletroquímica (EIS); cyclic voltammetry (CV).

**Table 4 biosensors-12-00572-t004:** Toxicological data established by USEPA regarding pesticides detected by biosensors.

Pesticides	ARfD (mg/kg bw)	ADI (mg/kg)	LD_50_ (mg/kg)
Acetamiprid	0.025	0.070	146
Fenpropathrin	0.061	0.030	870
Acetochlor	1.5	1.0	1929
Carbendazim	0.020	0.020	15,000
Carbaryl	0.10	0.20	303
Chlorpyrifos	0.011	0.12	500
Diazinon	0.00020	0.025	1160
Dichlorvos	0.10	0.0040	80
Dimethyl-methylphosphonate	-	-	5000
Ethylparathion	0.0050	0.00061	2.0
Fenthion	0.0010	0.0072	190
Isocarbophos	-	-	50
Malathion	0.30	0.030	1778
Methyl parathion	0.030	0.0030	3.0
Monocrotophos	0.0020	0.00060	112
Dimethoate	-	0.013	240
Nitrofen	-	-	5000
Paraoxon	-	-	1800
Phosmet	0.045	0.010	113
Pirimicarb	0.11	0.035	142
Pirimiphos methyl	0.15	0.004	1250
Profenofos	1.0	0.030	450
Triazophos	0.0012	0.00020	500

**Legend:** (-) not reported, (ARfD)—Acute reference Doses, (ADI) Acceptable daily intake, (LD_50_)—Median Lethal Dose.

## Data Availability

Not applicable.

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
