# Peer review of "Recent Advances in Nanomaterial-Based Biosensors for Pesticide Detection in Foods"

_biosensors, 2022, doi:10.3390/bios12080572_

Round 1

Reviewer 1 Report

In this work, the authors summarized the recent advances in nanomaterial-based biosensors for pesticide detection in foods. Excessive pesticide use relates to a variety of environmental issues, including eutrophication of water sources, high levels of pesticide residues in food, soil and water. Thus, design and development of high-performance biosensors is of greatly desirable. This review discussed different kinds of namaterials to sensitively and selectively detect target pesticide, and thus was very interesting. In this context, I recommend its publication in this work after the minor revision.

1. Up to now, there have been a great number of reviews on pesticides sensing reported. The difference of this work from other reviews should be described. In addition, why to summarize nanomaterial-based biosensor for pesticide?

2. Many nanomaterials have been applied to detect pesticide, the authors only discussed AuNMs, AgNMs, CNTs. This is not sufficient.

3. What is the train of thought in this work? 

4. The authors introduced many nanomaterial-based biosensors. However, what about the difference between them? For example, in 3.1.1.

5. To give readers a comprehensive understand of this review, some recent progress including Anal. Chem., 2021, 4084 should be emphatically described in detail in this work.

Author Response

Dear editor Dr. Zhongyu Cai,

Please find enclosed the revised manuscript entitled Recent advances in nanomaterial-based biosensors for pesticide detection in foods”.

We would like to thank reviewers who have performed critical and constructive work. We have addressed their criticisms, clarifying the issues raised and implementing all suggestions to the original text. All changes are highlighted in red in the manuscript. 

Best regards,

Carlos Adam Conte-Junior and coauthors 

Federal University of Rio de Janeiro

Reviewer 1

In this work, the authors summarized the recent advances in nanomaterial-based biosensors for pesticide detection in foods. Excessive pesticide use relates to a variety of environmental issues, including eutrophication of water sources, high levels of pesticide residues in food, soil and water. Thus, design and development of high-performance biosensors is of greatly desirable. This review discussed different kinds of namaterials to sensitively and selectively detect target pesticide, and thus was very interesting. In this context, I recommend its publication in this work after the minor revision.

Answer: Dear reviewer, we are happy that you liked the theme proposed in our work. We appreciate your availability to evaluate and for the significant contributions raised. Below, we responded to each of your comments.

  1. Up to now, there have been a great number of reviews on pesticides sensing reported. The difference of this work from other reviews should be described. In addition, why to summarize nanomaterial-based biosensor for pesticide?

Answer: Thanks for the observation. The differential of our work regarding that already published in the literature, as well as the cause of the focus on nanomaterials were clarified and are on page 3.

  1. Many nanomaterials have been applied to detect pesticide, the authors only discussed AuNMs, AgNMs, CNTs. This is not sufficient.

Answer: Thanks for the remark. New sections with other nanomaterials were written: Quantum Dots, Graphene and Reduced Graphene Oxide, and Titanium nanoparticles are marked in red.

  1. What is the train of thought in this work? 

Answer: Dear Reviewer, for better clarification, we have improved the section describing the scope/objective of this review as suggested. However, it is worth noting that the main difference lies in the systematic search for the articles that make up the review, since they are chosen based on eligibility criteria. In this case, it was the analytical performance of biosensors together with nanomaterials and biorecognition materials used in the construction of devices for the detection of pesticides in food. We hope the following changes, highlighted on page 4, have answered your question.

  1. The authors introduced many nanomaterial-based biosensors. However, what about the difference between them? For example, in 3.1.1.

Answer: Thanks for the observation. The difference between them was described on page 10.

“ While selecting articles to compose the systematic review, different types of nano-materials associated with varying types of biorecognition materials and other detection systems were observed. Some systems are applied from colorimetric to electrochemical de-tection, depending on the properties of the nanomaterial. SWCNTs and MWCNTs, for ex-ample, exhibit excellent thermal conductivities. Graphene has a high surface area com-pared to CNTs. Graphene oxide (GO), whose electronic conductivity is less. Carbon-based quantum dots have unique characteristics that make them extraordinary materials for di-verse applications, such as photoluminescence properties, biocompatibility, and low tox-icity. Metal-based nanomaterials Au or Ag, for example, have a high surface area and have the excellent adsorption ability of small molecules. These properties are associated with the low detection limits obtained with biosensors. Below are listed the main nano-materials used to manufacture biosensors for the detection of pesticides.”

  1. To give readers a comprehensive understand of this review, some recent progress including Anal. Chem., 2021, 4084 should be emphatically described in detail in this work.

Answer: As suggested by the reviewer we have inserted the sensors based on nanozyme in the section of “3.4. Highlights and future perspectives”. See the information entered: “Sensors based on nanozyme, which unify characteristics of a nanomaterial and a bioenzyme, have been the solution to solving the obstacles encountered in using natural enzymes. However, this concept still has some mishaps, as it is based mainly on a colorimetric technique, susceptible to severe color interference caused by the nanozymes and samples themselves. To overcome color interference, several efforts have been made in re-cent years. Recently, a promising electrochemical sensor based on nanozymes with manganese dioxide nanosheets (MnNS) for the detection of organophosphate pesticides (OPs) was proposed by Wu et al. [ref]. According to the authors, the MnNS eliminated color interference and allowed the detection of paraoxon in the pakchoi sample with high precision and good accuracy, with a LOD of 0.025 ng mL-1.”

Reviewer 2 Report

The authors have prepared a comprehensive review of the recent literature on the development of biosensors for pesticide detection. I have to recommend a major revision of the text with the following comments:

1. The authors must include citations top the recent reviews that were published on the same topic. For example, the following citation should have been made and compared with this submission so that the overlaps of same information are avoided. The authors should check the literature carefully for these existing reviews and acknowledge their existence with citations.

Recent advances in nanomaterials-based electrochemical (bio)sensors for pesticides detection

TrAC Trends in Analytical Chemistry

Volume 132, November 2020, 116041

2. In Table 3, the "Amperometric" biosensor is an "electrochemical" one. The authors should not categorize "Amperometric" as different from electrochemical. The authors should also clarify the electrochemical ones as which technique was used for the detection. Was it differential pulse voltammetry, square-wave voltammetry, cyclic voltammetry, electrochemical impedance spectroscopy, etc.?

3. The authors should also talk about the toxic dose or lethal dose of these pesticides allowed by the Environmental Protection Agency, European Union, or other World Health Organizations. Do these biosensors meet the requirements or fail to meet them to detect the pesticides? This information could be given ina Table or text.

4. There are no surface plasmon resonance or localized SPR biosensors mentioned in this review. The authors should also include SPR-based biosensors in this review.

5. English language must be checked by a native English speaker. for example "3.1.4. Hybrid nanoestructures" should be "3.1.4. Hybrid nanostructures" and "3.4. Highlights and futures perspectives" should be "3.4. Highlights and future perspectives". There are many more typos and grammatical errors in the text.

In view of my comments above, I would recommend a major revision.

Author Response

Dear editor Dr. Zhongyu Cai,

Please find enclosed the revised manuscript entitled Recent advances in nanomaterial-based biosensors for pesticide detection in foods”.

We would like to thank reviewers who have performed critical and constructive work. We have addressed their criticisms, clarifying the issues raised and implementing all suggestions to the original text. All changes are highlighted in red in the manuscript. 

Best regards,

Carlos Adam Conte-Junior and coauthors 

Federal University of Rio de Janeiro

Reviewer 2

The authors have prepared a comprehensive review of the recent literature on the development of biosensors for pesticide detection. I have to recommend a major revision of the text with the following comments:

Answer: Dear reviewer, we are happy that you liked the theme proposed in our work. We appreciate your availability to evaluate and for the significant contributions raised. Below, we responded to each of your comments.

  1. The authors must include citations top the recent reviews that were published on the same topic. For example, the following citation should have been made and compared with this submission so that the overlaps of same information are avoided. The authors should check the literature carefully for these existing reviews and acknowledge their existence with citations.

Recent advances in nanomaterials-based electrochemical (bio)sensors for pesticides detection-

TrAC Trends in Analytical Chemistry, Volume 132, November 2020, 116041

Answer: Thank you for the observation. This work and other recent publications have been included. We emphasize the differential of our work regarding those already published (pg 3) emphasizing that the systematic search methodology in the main databases allowed us to analyze all the recently published experienced studies. With this we can observe which types of biosensors were actually developed and applied in food samples.

  1. In Table 3, the "Amperometric" biosensor is an "electrochemical" one. The authors should not categorize "Amperometric" as different from electrochemical. The authors should also clarify the electrochemical ones as which technique was used for the detection. Was it differential pulse voltammetry, square-wave voltammetry, cyclic voltammetry, electrochemical impedance spectroscopy, etc.?

Answer. Thank you for your observation. The mentioned study was corrected, as well as all categorized as electrochemical in table 3.

  1. The authors should also talk about the toxic dose or lethal dose of these pesticides allowed by the Environmental Protection Agency, European Union, or other World Health Organizations. Do these biosensors meet the requirements or fail to meet them to detect the pesticides? This information could be given ina Table or text.

Answer: Thanks for the observation. A new section was written, (3.5. Efficiency of biosensors in legislation: MRL, ARfD and DL50) describing point-to-point suggestions and also table 4 was inserted. The modification is on page 30 marked in red.

  1. There are no surface plasmon resonance or localized SPR biosensors mentioned in this review. The authors should also include SPR-based biosensors in this review.

Answer: The surface plasmonic resonance (SPR) phenomenon is related to the resonant oscillation that causes increased absorption or index of refraction of a specific wavelength on the surface of the nanoparticles. These phenomena have been associated with detections in different biosensors, such as LSPR and enhanced Raman surface spectroscopic detection (SERS). Among the SERS and SPR detection systems, only sers detection, was observed among the selected articles after the systematized search and selection. LSPR has a limitation that is related to limited detection, and for the detection of pesticides in different samples, it is necessary to use sensitive systems with low detection limits (reference: doi: 10.3390/s150715684)

  1. English language must be checked by a native English speaker. for example "3.1.4. Hybrid nanoestructures" should be "3.1.4. Hybrid nanostructures" and "3.4. Highlights and futures perspectives" should be "3.4. Highlights and future perspectives". There are many more typos and grammatical errors in the text.

Answer. Thank you for your observation. The grammar of the manuscript was reviewed by a native.

In view of my comments above, I would recommend a major revision.

Reviewer 3 Report

This manuscript has made a review on nanomaterial-based biosensors for pesticide detection in foods. The topic itself is very valuable, however the organization of the article was still need to be improved. Here gives some suggestions for further revision:

1. Introduction section, combining the paragraphs 1-3 into one paragraph.

2. The review does not adequately summarize the most commonly utilized techniques to detect these pollutants.

3. The table should be present as 3 line table, such as Table 2 and Table 3.

4. “3.4. Highlights and futures perspectives” section should be strengthened.

5. Some figures should be added in “3.1.1 Gold Nanomaterials, 3.1.2 Silver Nanomaterials, 3.1.3 Carbon nanotubes” section.

Author Response

Dear editor Dr. Zhongyu Cai,

Please find enclosed the revised manuscript entitled Recent advances in nanomaterial-based biosensors for pesticide detection in foods”.

We would like to thank reviewers who have performed critical and constructive work. We have addressed their criticisms, clarifying the issues raised and implementing all suggestions to the original text. All changes are highlighted in red in the manuscript. 

Best regards,

Carlos Adam Conte-Junior and coauthors 

Federal University of Rio de Janeiro

Reviewer 3

This manuscript has made a review on nanomaterial-based biosensors for pesticide detection in foods. The topic itself is very valuable, however the organization of the article was still need to be improved. Here gives some suggestions for further revision:

Answer: Dear reviewer, we are happy that you liked the theme proposed in our work. We appreciate your availability to evaluate and for the significant contributions raised. Below, we responded to each of your comments.

  1. Introduction section, combining the “paragraphs 1-3” into one paragraph.

Answer. Thank you for your observation. The paragraphs have been combined as suggested and are on page 2 marked in red.

  1. The review does not adequately summarize the most commonly utilized techniques to detect these pollutants.

Answer: Thank you. The techniques used to detect this pollutant have been described in "Introducion", pg 2.

“Pesticides can be analyzed using different techniques class of separation and detection like gas chromatography (GC) with electron capture detection, flame ionization detection, nitrogen-phosphorus detection, mass spectrometry, and/or liquid chromatography (LC) with ultraviolet, diode array, fluorescence, or electrochemical detection and mass spectrometry.”

  1. The table should be present as 3 line table, such as Table 2 and Table 3.

Answer: Dear reviewer, it was not clear what the suggested changes would be to this question. It would be possible to give a better context so that we can implement such required corrections.

  1. “3.4. Highlights and futures perspectives” section should be strengthened.

Answer: Thanks for the observation. The section has been modified and is marked in red.

  1. Some figures should be added in “3.1.1 Gold Nanomaterials, 3.1.2 Silver Nanomaterials, 3.1.3 Carbon nanotubes” section.

Answer: Thank you. New figures have been added in the sections mentioned (figure 4, 5 and 6) as well as in new sections (Figure 7 and 8).

Round 2

Reviewer 3 Report

Authors have answered my questions in my previous comments. Please see below suggestions for this manuscript version.

1. The format of Table 2 and Table 3 should be present as three line table, like Table 1 and Table 4.